# DEBIASED GRAPH NEURAL NETWORKS WITH AGNOSTIC LABEL SELECTION BIAS

## ABSTRACT

Most existing Graph Neural Networks (GNNs) are proposed without considering the selection bias in data, i.e., the inconsistent distribution between the training set with test set. In reality, the test data is not even available during the training process, making selection bias agnostic. Training GNNs with biased selected nodes leads to significant parameter estimation bias and greatly impacts the generalization ability on test nodes. In this paper, we first present an experimental investigation, which clearly shows that the selection bias drastically hinders the generalization ability of GNNs, and theoretically prove that the selection bias will cause the biased estimation on GNN parameters. Then to remove the bias in GNN estimation, we propose a novel Debiased Graph Neural Networks (DGNN) with a differentiated decorrelation regularizer. The differentiated decorrelation regularizer estimates a sample weight for each labeled node such that the spurious correlation of learned embeddings could be eliminated. We analyze the regularizer in causal view and it motivates us to differentiate the weights of the variables based on their contribution on the confounding bias. Then, these sample weights are used for reweighting GNNs to eliminate the estimation bias, thus help to improve the stability of prediction on unknown test nodes. Comprehensive experiments are conducted on several challenging graph datasets with two kinds of label selection bias. The results well verify that our proposed model outperforms the state-of-the-art methods and DGNN is a flexible framework to enhance existing GNNs.

## 1 INTRODUCTION

Graph Neural Networks (GNNs) are powerful deep learning algorithms on graphs with various applications (Scarselli et al., 2008; Kipf & Welling, 2016; Veličković et al., 2017; Hamilton et al., 2017). Existing GNNs mainly learn a node embedding through aggregating the features from its neighbors, and such message-passing framework is supervised by node label in an end-to-end manner. During this training procedure, GNNs will effectively learn the correlation between the structure pattern and node feature with node label, so that GNNs are capable of learning the embeddings of new nodes and inferring their labels.

One basic requirement of GNNs making precise prediction on unseen test nodes is that the distribution of labeled training and test nodes is same, i.e., the structure and feature of labeled training and test nodes follow the similar pattern, so that the learned correlation between the current graph and label can be well generalized to the new nodes. However, in reality, there are two inevitable issues. (1) Because it is difficult to control the graph collection in an unbiased environment, the relationship between the collected real-world graph and the labeled nodes is inevitably biased. Training on such graph will cause biased correlation with node label. Taking a scientist collaboration network as an example, if most scientists with "machine learning" (ML) label collaborate with those with "computer vision" (CV) label, existing GNNs may learn spurious correlation, i.e., scientists who cooperate with CV scientist are ML scientists. If a new ML scientist only connects with ML scientists or the scientists in other areas, it will be probably misclassified. (2) The test node in the real scenario is usually not available, implying that the distribution of new nodes is agnostic. Once the distribution is inconsistent with that in the training nodes, the performance of all the current GNNs will be hindered. Even transfer learning is able to solve the distribution shift problem, however, it still needs the prior of test distribution, which actually cannot be obtained beforehand. Therefore, the agnostic label selection bias greatly affects the generalization ability of GNNs on unknown test data.

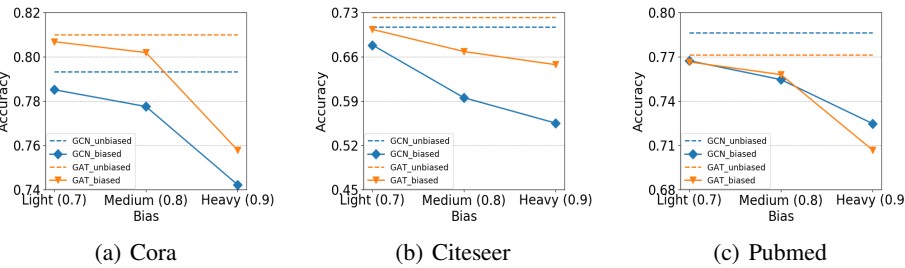

Figure 1: Effect of selection bias on GCN and GAT.

In order to observe selection bias in real graph data, we conduct an experimental investigation to validate the effect of selection bias on GNNs (details can be seen in Section 2.1). We select training nodes with different biased degrees for each dataset, making the distribution of training nodes and test nodes inconsistent. The results clearly show that selection bias drastically hinders the performance of GNNs on unseen test nodes. Moreover, with heavier bias, the performance drops more. Further, we theoretically analyze how the data selection bias results in the estimation bias in GNN parameters (details can be seen in Section 2.2). Based on the stable learning technique (Kuang et al., 2020), we can assume that the learned embeddings consist of two parts: stable variables and unstable variables. The data selection bias will cause the spurious correlation between these two kinds of variables. Thereby we prove that with the inevitable model misspecification, the spurious correlation will further cause the parameter estimation bias. Once the weakness of the current GNNs with selection bias is identified, one natural question is "*how to remove the estimation bias in GNNs?*"

In this paper, we propose a novel Debiased Graph Neural Network (DGNN) framework for stable graph learning by jointly optimizing a differentiated decorrelation regularizer and a weighted GNN model. Specifically, the differentiated decorrelation regularizer is able to learn a set of sample weights under differentiated variable weights, so that the spurious correlation between stable and unstable variables would be greatly eliminated. Based on the causal view analysis of decorrelation regularizer, we theoretically prove that the weights of variables can be differentiated by the regression weights. Moreover, to better combine the decorrelation regularizer with GNNs, we prove that adding the regularizer to the embedding learned by the second to last layer could be both theoretically sound and flexible. Then the sample weights learned by decorrelation regularizer are used to reweight GNN loss so that the parameter estimation could be unbiased.

In summary, the contributions of this paper are three-fold: i) We investigate a new problem of learning GNNs with agnostic label selection bias. The problem setting is general and practical for real applications. ii) We bring the idea of variable decorrelation into GNNs to relieve bias influence on model learning and propose a general framework DGNN which could be adopted to various GNNs. iii) We conduct the experiments on real-world graph benchmarks with two kinds of agnostic label selection bias, and the experimental results demonstrate the effectiveness and flexibility of our model.

## 2 EFFECT OF LABEL SELECTION BIAS ON GNNS

In this section, we first formulate our target problem as follows:

**Problem 1** (**Semi-supervised Learning on Graph with Agnostic Label Selection Bias**). *Given a training graph $\mathcal{G}_{train} = \{\mathbf{A}_{train}, \mathbf{X}_{train}, \mathbf{Y}_{train}\}$, where $\mathbf{A}_{train} \in \mathbb{R}^{N \times N}$ (N nodes) represents the adjacency matrix, $\mathbf{X}_{train} \in \mathbb{R}^{N \times D}$ (D features) refers to the node features and $\mathbf{Y}_{train} \in \mathbb{R}^{n \times C}$ (n labeled nodes, C classes) refers to the available labels for training ($n \ll N$), the task is to learn a GNN $g_\theta(\cdot)$ with parameter $\theta$ to precisely predict the label of nodes on test graph $\mathcal{G}_{test} = \{\mathbf{A}_{test}, \mathbf{X}_{test}, \mathbf{Y}_{test}\}$, where distribution $\Psi(\mathcal{G}_{train}) \neq \Psi(\mathcal{G}_{test})$.*

### 2.1 EXPERIMENTAL INVESTIGATION

We conduct an experimental investigation to examine whether the state-of-the-art GNNs are sensitive to the selection bias. The main idea is that we will perform two representative GNNs: GCN (Kipf

& Welling, 2016) and GAT (Veličković et al., 2017) on three widely used graph datasets: *Cora*, *Citeseer*, *Pubmed* (Sen et al., 2008) with different degrees of bias. If the performance drops sharply in comparison with the scenarios without selection bias, this will demonstrate that GNNs cannot generalize well in selection bias setting.

To simulate the agnostic selection bias scenario, we first follow the inductive setting in Wu et al. (2019) that masks the validation and test nodes as the training graph $\mathcal{G}_{train}$ in the training phase, and then infer the labels of validation and test nodes with whole graph $\mathcal{G}_{test}$. In this way, the distribution of test node can be considered agnostic. Following Zadrozny (2004), we design a biased label selection method on training graph $\mathcal{G}_{train}$. The selection variable $e$ is introduced to control whether the node will be selected as labeled nodes, where $e = 1$ means selected and 0 otherwise. For node $i$, we compute its neighbor distribution ratio: $r_i = |\{j|j \in \mathcal{N}_i, y_j \neq y_i\}|/|\mathcal{N}_i|$, where $\mathcal{N}_i$ is neighborhood of node $i$ in $\mathcal{G}_{train}$ and $y_j \neq y_i$ means the label of central node $i$ is not the label of its neighborhood node $j$. And $r_i$ measures the difference between the label of central node $i$ with the labels of its neighborhood. Then we average all the nodes' $r$ to get a threshold $t$. For each node, the probability to be selected is: $P(e_i = 1|r_i) = \left\{ \begin{array}{ll} \epsilon & r_i \geq t \\ 1 - \epsilon & r_i < t \end{array} \right.$ , where $\epsilon \in (0.5, 1)$ is used to control the degree of selection bias and the larger $\epsilon$ means heavier bias. We set $\epsilon$ as $\{0.7, 0.8, 0.9\}$ to get three bias degrees for each dataset, termed as *Light, Medium, Heavy*, respectively. We select 20 nodes for each class for training and the validation and test nodes are same as Yang et al. (2016). Furthermore, we take the *unbiased* datasets as baselines, where the labeled nodes are selected randomly.

Figure 1 is the results of GCN and GAT on biased datasets. The dashed lines mean the performances of GCN/GAT on unbiased datasets and the solid lines refer to the results on biased datasets. We can find that: i) The dashed lines are all above the corresponding coloured solid lines, indicating that selection bias greatly affects the GNNs' performance. ii) All solid lines decrease monotonically with the increase of bias degree, demonstrating that heavier bias will cause larger performance decrease.

## 2.2 THEORETICAL ANALYSIS

The above experiment empirically verifies the effect of selection bias on GNNs. Here we theoretically analyze the effect of selection bias on estimating the parameters in GNNs. First, because biased labeled nodes have biased neighborhood structure, GNNs will encode this biased information into the node embeddings. Based on stable learning technique (Kuang et al., 2020), we make following assumption:

**Assumption 1.** *All the variables of embeddings learned by GNNs for each node can be decomposed as* $\mathbf{H} = \{\mathbf{S}, \mathbf{V}\}$, *where* $\mathbf{S}$ *represents the stable variables and* $\mathbf{V}$ *represents the unstable variables. Specifically, for both training and test environment,* $\mathbb{E}(\mathbf{Y}|\mathbf{S} = s, \mathbf{V} = v) = \mathbb{E}(\mathbf{Y}|\mathbf{S} = s)$.

Under Assumption 1, the distribution shift between training set and test set is mainly induced by the variation in the joint distribution over $(\mathbf{S}, \mathbf{V})$, i.e., $\mathbb{P}(\mathbf{S}_{train}, \mathbf{V}_{train}) \neq \mathbb{P}(\mathbf{S}_{test}, \mathbf{V}_{test})$. However, there is an invariant relationship between stable variable $\mathbf{S}$ and outcome $\mathbf{Y}$ in both training and test environments, which can be expressed as $\mathbb{P}(\mathbf{Y}_{train}|\mathbf{S}_{train}) = \mathbb{P}(\mathbf{Y}_{test}|\mathbf{S}_{test})$. Assumption 1 can be guaranteed by $\mathbf{Y} \perp \mathbf{V}|\mathbf{S}$. Thus, one can solve the stable prediction problem by developing a function $f(\cdot)$ based on $\mathbf{S}$. However, one can hardly identify such variables in GNNs.

Without loss of generality, we take $\mathbf{Y}$ as continuous variable for analysis and have the following assumption:

**Assumption 2.** *The true generation process of target variable* $\mathbf{Y}$ *contains not only the linear combination of stable variables* $\mathbf{S}$, *but also the nonlinear transformation of stable variables.*

Based on the above assumptions, we formalize the label generation process as follows:

$$\mathbf{Y} = f(\mathbf{X}, \mathbf{A}) + \varepsilon = \mathscr{G}(\mathbf{X}, \mathbf{A}; \theta_g)_S \beta_S + \mathscr{G}(\mathbf{X}, \mathbf{A}; \theta_g)_V \beta_V + g(\mathscr{G}(\mathbf{X}, \mathbf{A}; \theta_g)_S) + \varepsilon, \quad (1)$$

where $\mathscr{G}(\mathbf{X}, \mathbf{A}; \theta_g) \in \mathbb{R}^{N \times p}$ denotes an unknown function of $\mathbf{X}$ and $\mathbf{A}$ that learns node embedding and it can be learned by a GNN, such as GCN and GAT, the output variables of $\mathscr{G}(\mathbf{X}, \mathbf{A}; \theta_g)$ can be decomposed as stable variables $\mathscr{G}(\mathbf{X}, \mathbf{A}; \theta_g)_S \in \mathbf{R}^{N \times m}$ and unstable variables $\mathscr{G}(\mathbf{X}, \mathbf{A}; \theta_g)_V \in \mathbf{R}^{N \times q}$ $(m + q = p)$, $\beta_S \in \mathbb{R}^{m \times 1}$ and $\beta_V \in \mathbb{R}^{q \times 1}$ are the linear coefficients can be learned by the last layer of GNNs, $\varepsilon$ is the independent random noise, and $g(\cdot)$ is the nonlinear transformation function

of stable variables. According to Assumption 1, we know that coefficients of unstable variables $\mathscr{G}(\mathbf{X}, \mathbf{A}; \theta_g)_V$ are actually 0 (i.e., $\beta_V = 0$).

For a classical GNN model with linear regressor, its prediction function can be formulated as:

$$\hat{\mathbf{Y}} = \hat{\mathscr{G}}(\mathbf{X}, \mathbf{A}; \theta_g)_S \hat{\beta}_S + \hat{\mathscr{G}}(\mathbf{X}, \mathbf{A}; \theta_g)_V \hat{\beta}_V + \varepsilon. \tag{2}$$

Compared with Eq. (1), we can find that the parameters of GNN could be unbiasedly estimated if the nonlinear term $g(\mathscr{G}(\mathbf{X}, \mathbf{A}; \theta_g)_S) = 0$, because the GNN model will have the same label generation mechanism as Eq. (1). However, limited by the nonlinear power of GNNs (Xu et al., 2019), it is reasonable to assume that there is a nonlinear term $g(\mathscr{G}(\mathbf{X}, \mathbf{A}; \theta_g)_S) \neq 0$ that cannot be fitted by the GNNs. Under this assumption, next, we taking a vanilla GCN (Kipf & Welling, 2016) as an example to illustrate how the distribution shift will induce parameter estimation bias. A two-layer GCN can be formulated as $\hat{\mathbf{A}}\sigma(\hat{\mathbf{A}}\mathbf{X}\mathbf{W}^{(0)})\mathbf{W}^{(1)}$, where $\hat{\mathbf{A}}$ is the normalized adjacency matrix, $\mathbf{W}$ is the transformation matrix at each layer and $\sigma(\cdot)$ is the Relu activation function. We decompose GCN as two parts: one is embedding learning part $\hat{\mathbf{A}}\sigma(\hat{\mathbf{A}}\mathbf{X}\mathbf{W}^{(0)})$, which can be decomposed as $[\mathbf{S}^{\mathrm{T}}, \mathbf{V}^{\mathrm{T}}]$, corresponding to $\hat{\mathscr{G}}(\mathbf{X}, \mathbf{A}; \theta_g)_S$ and $\hat{\mathscr{G}}(\mathbf{X}, \mathbf{A}; \theta_g)_V$ in Eq. (2), and the other part is $\mathbf{W}^{(1)}$, where the learned parameters can be decomposed as $[\tilde{\beta}_S, \tilde{\beta}_V]$, corresponding to $[\hat{\beta}_S, \hat{\beta}_V]$ in Eq. (2). We aim at minimizing the square loss: $\mathcal{L}_{GCN} = \sum_{i=1}^{n}(\mathbf{S}_i^{\mathrm{T}}\tilde{\beta}_S + \mathbf{V}_i^{\mathrm{T}}\tilde{\beta}_V - \mathbf{Y}_i)^2$.

According to the derivation rule of partitioned regression model, we have:

$$\tilde{\beta}_V - \beta_V = (\frac{1}{n}\sum_{i=1}^{n}\mathbf{V}_i^{\mathrm{T}}\mathbf{V}_i)^{-1}(\frac{1}{n}\sum_{i=1}^{n}\mathbf{V}_i^{\mathrm{T}}g(\mathbf{S}_i)) + (\frac{1}{n}\sum_{i=1}^{n}\mathbf{V}_i^{\mathrm{T}}\mathbf{V}_i)^{-1}(\frac{1}{n}\sum_{i=1}^{n}\mathbf{V}_i^{\mathrm{T}}\mathbf{S}_i)(\beta_S - \tilde{\beta}_S), \tag{3}$$

$$\tilde{\beta}_S - \beta_S = (\frac{1}{n}\sum_{i=1}^{n}\mathbf{S}_i^{\mathrm{T}}\mathbf{S}_i)^{-1}(\frac{1}{n}\sum_{i=1}^{n}\mathbf{S}_i^{\mathrm{T}}g(\mathbf{S}_i)) + (\frac{1}{n}\sum_{i=1}^{n}\mathbf{S}_i^{\mathrm{T}}\mathbf{S}_i)^{-1}(\frac{1}{n}\sum_{i=1}^{n}\mathbf{S}_i^{\mathrm{T}}\mathbf{V}_i)(\beta_V - \tilde{\beta}_V), \tag{4}$$

where $n$ is labeled node size, $\mathbf{S}_i$ is $i$-th sample of $\mathbf{S}$, $\frac{1}{n}\sum_{i=1}^{n}\mathbf{V}_i^{\mathrm{T}}g(\mathbf{S}_i) = \mathbb{E}(\mathbf{V}^{\mathrm{T}}g(\mathbf{S})) + o_p(1)$, $\frac{1}{n}\sum_{i=1}^{n}\mathbf{V}_i^{\mathrm{T}}\mathbf{S}_i = \mathbb{E}(\mathbf{V}^{\mathrm{T}}\mathbf{S}) + o_p(1)$ and $o_p(1)$ is the error which is negligible. Ideally, $\tilde{\beta}_V - \beta_V = 0$ indicates that there is no bias between the estimated and the real parameter. However, if $\mathbb{E}(\mathbf{V}^{\mathrm{T}}\mathbf{S}) \neq 0$ or $\mathbb{E}(\mathbf{V}^{\mathrm{T}}g(\mathbf{S})) \neq 0$ in Eq. (3), $\tilde{\beta}_V$ will be biased, leading to the biased estimation on $\tilde{\beta}_S$ in Eq. (4) as well. Since the correlation between $\mathbf{V}$ and $\mathbf{S}$ (or $g(\mathbf{S})$) might shift in test phase, the biased parameters learned in training set is not the optimal parameters for predicting testing nodes. Therefore, to increase the stability of prediction, we need to unbiasedly estimate the parameters of $\tilde{\beta}_V$ by removing the correlation between $\mathbf{V}$ and $\mathbf{S}$ (or $g(\mathbf{S})$) on training graph, making $\mathbb{E}(\mathbf{V}^{\mathrm{T}}\mathbf{S}) = 0$ or $\mathbb{E}(\mathbf{V}^{\mathrm{T}}g(\mathbf{S})) = 0$. Note that $\frac{1}{n}\sum_{i=1}^{n}\mathbf{S}_i^{\mathrm{T}}g(\mathbf{S}_i)$ in Eq. (4) can also cause estimation bias, but the relation between $\mathbf{S}$ and $g(\mathbf{S})$ is stable across environments, which do not influence the stability to some extent.

## 3 PROPOSED MODEL

### 3.1 REVISITING ON VARIABLE DECORRELATION IN CAUSAL VIEW

To decorrelate $\mathbf{V}$ and $\mathbf{S}$ (or $g(\mathbf{S})$), we should decorrelate the output variables of $\hat{\mathscr{G}}(\mathbf{X}, \mathbf{A}; \theta_g)$ (Kuang et al., 2020). They propose a Variable Decorrelation (VD) term with sample reweighting technique to eliminate the correlation between each variable pair, in which the sample weights are learned by jointly minimizing the moment discrepancy between each variable pair:

$$\mathcal{L}_{VD}(\mathbf{H}) = \sum_{j=1}^{p} ||\mathbf{H}_{\cdot j}^{\mathrm{T}}\Lambda_{\mathbf{w}}\mathbf{H}_{\cdot -j}/n - \mathbf{H}_{\cdot j}^{\mathrm{T}}\mathbf{w}/n \cdot \mathbf{H}_{\cdot -j}^{\mathrm{T}}\mathbf{w}/n||_2^2, \tag{5}$$

where $\mathbf{H} \in \mathbb{R}^{n \times p}$ means the variables needed to be decorrelated, $\mathbf{H}_{\cdot j}$ is $j$-th variable of $\mathbf{H}$, $\mathbf{H}_{\cdot -j} = \mathbf{H} \backslash \mathbf{H}_{\cdot j}$ means all the remaining variables by setting the value of $j$-th variable in $\mathbf{H}$ as *zero*, $\mathbf{w} \in \mathbb{R}^{n \times 1}$ are sample weights, $\sum_{i=1}^{n}\mathbf{w}_i = n$ and $\Lambda_{\mathbf{w}} = \mathrm{diag}(\mathbf{w}_1, \cdots, \mathbf{w}_n)$ is the corresponding diagonal matrix. As we can see, $\mathcal{L}_{VD}(\mathbf{H})$ can be reformulated as $\sum_{j \neq k} ||\mathbf{H}_{\cdot j}^{\mathrm{T}}\Lambda_{\mathbf{w}}\mathbf{H}_{\cdot k}/n - \mathbf{H}_{\cdot j}^{\mathrm{T}}\mathbf{w}/n \cdot \mathbf{H}_{\cdot k}^{\mathrm{T}}\mathbf{w}/n||_2^2$, and it aims to let $\mathbb{E}(\mathbf{H}_{\cdot i}^{\mathrm{T}}\mathbf{H}_{\cdot j}) = \mathbb{E}(\mathbf{H}_{\cdot i}^{\mathrm{T}})\mathbb{E}(\mathbf{H}_{\cdot j})$ for each variable pair $j$ and $k$. $\mathcal{L}_{VD}(\mathbf{H})$ decorrelates all

the variable pairs equally. However, decorrelating all the variables requires sufficient samples Kuang et al. (2020), i.e., $n \rightarrow \infty$, which is hard to be satisfied, especially in the semi-supervised setting. In this scenario, we cannot guarantee $\mathcal{L}_{VD}(\mathbf{H}) = 0$. Therefore the key challenge is how to remove the correlation influencing the unbiased estimation most when $\mathcal{L}_{VD}(\mathbf{H}) \neq 0$.

Inspired by confounding balancing technique in observational studies (Hainmueller, 2012), we revisit the variable decorrelation regularizer in causal view and show how to differentiate each variable pair. Confounding balancing techniques are often used for causal effect estimation of treatment $T$, where the distributions of confounders $\mathbf{X}$ are different between treated ($T = 1$) and control ($T = 0$) groups because of non-random treatment assignment. One could balance the distribution of confounders between treatment and control groups to unbiased estimate causal treatment effects (Yao et al., 2020). Most balancing approaches exploit moments to characterize distributions, and balance them by adjusting sample weights $\mathbf{w}$ as follows: $\mathbf{w} = \arg\min_{\mathbf{w}} || \sum_{i:T_i=1} \mathbf{X}_i - \sum_{i:T_i=0} \mathbf{w}_i \cdot \mathbf{X}_i ||_2^2$. After balancing, the treatment $T$ and confounders $\mathbf{X}$ tend to be independent.

Given one targeted variable $j$, under the variables only have linear relation assumption[1], its decorrelation term, $\mathcal{L}_{VD_j} = ||\mathbf{H}_{\cdot j}^{\mathrm{T}} \Lambda_{\mathbf{w}} \mathbf{H}_{\cdot -j}/n - \mathbf{H}_{\cdot j}^{\mathrm{T}} \mathbf{w}/n \cdot \mathbf{H}_{\cdot -j}^{\mathrm{T}} \mathbf{w}/n ||_2^2$, is to make $\mathbf{H}_{\cdot j}$ independent of $\mathbf{H}_{\cdot -j}$, which is same as the confounding balancing term making treatment and confounders independent. Thereby, $\mathcal{L}_{VD_j}$ can also be viewed as a confounding balancing term, where $\mathbf{H}_{\cdot j}$ is treatment and $\mathbf{H}_{\cdot -j}$ is confounders, illustrated in Fig. 2(a). Hence, our target can be explained as unbiasedly estimate causal effect of each variable which is invariant across training and test set. As different variable may contribute unequally to the confounding bias, it is necessary to differentiate the confounders. The target of differentiating confounders exactly matches our target that removes the correlation of variables influencing the unbiased estimation most.

## 3.2 DIFFERETIATED VARIABLE DECORRELATION

Considering the continuous treatment, the causal effect of treatment can be measured by Marginal Treatment Effect Function (MTEF) (Kreif et al., 2015), and defined as: $MTEF = \frac{\mathbb{E}[Y_i(t)] - \mathbb{E}[Y_i(t-\Delta t)]}{\Delta t}$, where $Y_i(t)$ represents the potential outcome of sample $i$ with treatment status $T = t$, $\mathbb{E}(\cdot)$ refers to the expectation function, and $\Delta t$ denotes the increasing level of treatment. With the sample weights $\mathbf{w}$ decorrelating treatment and confounders, we can estimate the MTEF by:

$$\widehat{MTEF} = \frac{\sum_{i:T_i=t} \mathbf{w}_i \cdot Y_i(t) - \sum_{j:T_j=t-\Delta t} \mathbf{w}_j \cdot Y_j(t - \Delta t)}{\Delta t}. \tag{6}$$

Next we theoretically analyze how to differentiate confounders' weights with following theorem.

**Theorem 1.** *In observational studies, different confounders make unequal confounding bias on Marginal Treatment Effect Function (MTEF) with their own weights, and the weights can be learned via regressing outcome $Y$ on confounders $\mathbf{X}$ and treatment variable $T$.*

We prove Theorem 1 with the following assumption:

**Assumption 3** (Linearity). *The regression of outcome $Y$ on confounders $\mathbf{X}$ and treatment variable $T$ is linear, that is $Y = \sum_{k \neq t} \alpha_k \mathbf{X}_{\cdot k} + \alpha_t T + c + \varepsilon$, where $\alpha_k \in \alpha$ is the linear coefficient.*

Under Assumption 3, we can write estimator of $\widehat{MTEF}$ as:

$$\widehat{MTEF} = \frac{\sum_{i:T_i=t} \mathbf{w}_i \cdot Y_i(t) - \sum_{j:T_j=t-\Delta t} \mathbf{w}_j \cdot Y_j(t - \Delta t)}{\Delta t}$$
$$= MTEF + \sum_{k \neq t} \alpha_k \left( \frac{\sum_{i:T_i=t} \mathbf{w}_i \cdot \mathbf{X}_{ik} - \sum_{j:T_j=t-\Delta t} \mathbf{w}_j \cdot \mathbf{X}_{jk}}{\Delta t} \right) + \phi(\varepsilon), \tag{7}$$

where $MTEF$ is the ground truth, $\phi(\varepsilon)$ means the noise term, and $\phi(\varepsilon) \simeq 0$ with Gaussian noise. The detailed derivation can be found in Appendix A. To reduce the bias of $\widehat{MTEF}$, we need regulate the term $\sum_{k \neq t} \alpha_k \left( \frac{\sum_{i:T_i=t} \mathbf{w}_i \cdot \mathbf{X}_{ik} - \sum_{j:T_j=t-\Delta t} \mathbf{w}_j \cdot \mathbf{X}_{jk}}{\Delta t} \right)$, where $\frac{\sum_{i:T_i=t} \mathbf{w}_i \cdot \mathbf{X}_{ik} - \sum_{j:T_j=t-\Delta t} \mathbf{w}_j \cdot \mathbf{X}_{jk}}{\Delta t}$

---

[1]Nonlinear relation between variables can be incorporated by considering high-order moments in Eq. (5).

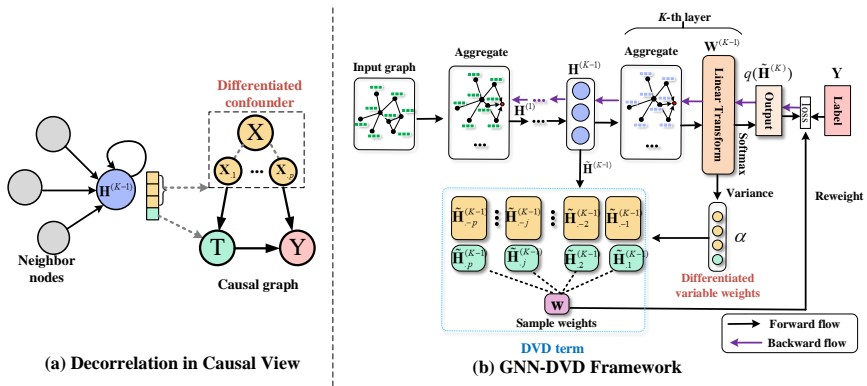

Figure 2: (a) Diagram of decorrelating node embedding with confounding balance. $\mathbf{H}^{(K-1)}$ is the node embedding to be decorrelated. $T$ is the treatment, corresponding to one target variable in $\mathbf{H}^{(K-1)}$. $\mathbf{X}$ is the confounders, corresponding to the remaining variables of the target variable in $\mathbf{H}^{(K-1)}$. $Y$ is the outcome, corresponding to labels. (b) The framework of GNN-DVD. The same color in the two figures represents the same kind of variable.

means the difference of the $k$-th confounder between treated and control samples. The parameter $\alpha_k$ represents the confounding bias weight of the $k$-th confounder, and it is the coefficient of $\mathbf{X}_{.k}$. Moreover, because our target is to learn the weight of each variable pair, i.e., between treatment and each confounder, we need to learn the weight $\alpha_t$ of treatment that is the coefficient of $T$. Hence, the confounder weights and treatment weight can be learned from the regression of observed outcome $Y$ on confounders $\mathbf{X}$ and treatment $T$ under *Linearity* assumption.

Due to the connection between treatment effect estimation with variable decorrelation as analyzed in Section 3.1, we utilize Theorem 1 to reweight the variable weight in variable decorrelation term. When apply the Theorem 1 to GNNs, the confounders $\mathbf{X}$ should be $\mathbf{H}_{.-j}$ and treatment is $\mathbf{H}_{.j}$, where the embedding $\mathbf{H}$ is learned by $\hat{\mathscr{G}}(\mathbf{X}, \mathbf{A}; \theta_g)$ in Eq. (2). And the variable weights $\alpha$ could be computed from the regression coefficients for $\mathbf{H}$, hence $\alpha$ is equal to $\hat{\beta}$ in Eq. (2). Then the Differentiated Variable Decorrelation (DVD) term can be formulated as follows:

$$
\begin{aligned}
\min_{\mathbf{w}} \mathcal{L}_{DVD}(\mathbf{H}) = \sum_{j=1}^{p} (\alpha^{\mathrm{T}} \cdot \mathrm{abs}(\mathbf{H}_{.j}^{\mathrm{T}} \Lambda_{\mathbf{w}} \mathbf{H}_{.-j}/n - \mathbf{H}_{.j}^{\mathrm{T}} \mathbf{w}/n \cdot \mathbf{H}_{.-j}^{\mathrm{T}} \mathbf{w}/n))^2 \\
+ \frac{\lambda_1}{n} \sum_{i=1}^{n} \mathbf{w}_i^2 + \lambda_2 (\frac{1}{n} \sum_{i=1}^{n} \mathbf{w}_i - 1)^2, s.t. \mathbf{w} \geq 0
\end{aligned}
\tag{8}
$$

where $\mathrm{abs}(\cdot)$ means the element-wise absolute value operation, preventing positive and negative values from eliminating. Term $\frac{\lambda_1}{n} \sum_{i=1}^{n} \mathbf{w}_i^2$ is added to reduce the variance of sample weights to achieve stability, and the formula $\lambda_2(\frac{1}{n} \sum_{i=1}^{n} \mathbf{w}_i - 1)^2$ avoids all the sample weights to be 0. The term $\mathbf{w} \geq 0$ constrains each sample weight to be non-negative. After variable reweighting, the weighted decorrelation term in Eq. (8) can be rewritten as $\sum_{j \neq k} \alpha_j^2 \alpha_k^2 ||\mathbf{H}_{.j}^{\mathrm{T}} \Lambda_{\mathbf{w}} \mathbf{H}_{.k}/n - \mathbf{H}_{.j}^{\mathrm{T}} \mathbf{w}/n \cdot \mathbf{H}_{.k}^{\mathrm{T}} \mathbf{w}/n||_2^2$, and the weight for variable pair $j$ and $k$ would be $\alpha_j^2 \alpha_k^2$, hence it considers both the weights of treatment and confounder. We prove the uniqueness property of $\mathbf{w}$ in Appendix B, as follows:

**Theorem 2** (Uniqueness). *If $\lambda_1 n \gg p^2 + \lambda_2$, $p^2 \gg \max(\lambda_1, \lambda_2)$, $|\mathbf{H}_{i,j}| \leq c$ and $|\alpha_i| \leq c$ for some constant $c$, the solution $\hat{\mathbf{w}} \in \{\mathbf{w} : |\mathbf{w}_i| \leq c\}$ to minimize Eq. (8) is unique.*

### 3.3 DEBIASED GNN FRAMEWORK

In this section, we describe the framework of Debiased GNN that incorporates DVD/VD term with GNNs in a seamless way. As analyzed in Section 2.2, decorrelating $\hat{\mathbf{A}} \sigma(\hat{\mathbf{A}} \mathbf{X} \mathbf{W}^{(0)})$ could make GCN stable. However, most GNNs follow a layer-by-layer stacking structure, and the output embedding of each layer is more easy to obtain in implementing. Since $\hat{\mathbf{A}} \sigma(\hat{\mathbf{A}} \mathbf{X} \mathbf{W}^{(0)})$ is the aggregation of

the first layer embedding $\sigma(\hat{\mathbf{A}}\mathbf{X}\mathbf{W}^{(0)})$, decorrelating these variables may lack the flexibility that incorporates DVD/VD term with other GNN structure. Fortunately, we have the following theorem to identify a more flexible way to combine variable decorrelation with GNNs.

**Theorem 3.** *Given $p$ pairwise uncorrelated variables $\mathbf{Z} = (\mathbf{Z}_1, \mathbf{Z}_2, \cdots, \mathbf{Z}_p)$, with a linear aggregation operator $\hat{\mathbf{A}}$, the variables of $\mathbf{Y} = \hat{\mathbf{A}}\mathbf{Z}$ are still pairwise uncorrelated.*

Proof can be found in Appendix C. The theorem indicates that if the variables of embeddings $\mathbf{Z}$ are uncorrelated, after any form of linear neighborhood aggregation $\hat{\mathbf{A}}$, e.g., average, attention or sum, the variables of transformed embeddings $\mathbf{Y}$ would be also uncorrelated. Therefore, decorrelating $\sigma(\hat{\mathbf{A}}\mathbf{X}\mathbf{W}^{(0)})$ can also reduce the estimation bias. For a $K$ layers of GNN, we can directly decorrelate the output of $(K-1)$-th layer, i.e., $\sigma(\hat{\mathbf{A}}\cdots\sigma(\hat{\mathbf{A}}\mathbf{X}\mathbf{W}^{(0)})\cdots\mathbf{W}^{(K-2)})$ for a $K$ layers of GCN.

The previous analysis finds a flexible way to incorporate DVD/VD term with GNNs, however, recall that we analyze GNNs based on the least squares loss, and most existing GNNs are designed for classification. Therefore, in the following, we analyze that the previous conclusions are still applicable in classification. We consider the cases that softmax layer is used as the output layer of GNNs and loss is the cross-entropy error function. We use the Newton-Raphson update rule (Bishop, 2006) to bridge the gap between linear regression and multi-classification. According to the Newton-Raphson update rule, the update formula for transformation matrix $\mathbf{W}^{(K-1)}$ of the last layer of GCN can be derived:

$$
\begin{aligned}
\mathbf{W}_{\cdot j}^{(\text{new})} &= \mathbf{W}_{\cdot j}^{(\text{old})} - (\mathbf{H}^{\mathrm{T}}\mathbf{R}\mathbf{H})^{-1}\mathbf{H}^{\mathrm{T}}(\mathbf{H}\mathbf{W}_{\cdot j}^{(\text{old})} - \mathbf{Y}_{\cdot j}) \\
&= (\mathbf{H}^{\mathrm{T}}\mathbf{R}\mathbf{H})^{-1}\{\mathbf{H}^{\mathrm{T}}\mathbf{R}\mathbf{H}\mathbf{W}_{\cdot j}^{(\text{old})} - \mathbf{H}^{\mathrm{T}}(\mathbf{H}\mathbf{W}_{\cdot j}^{(\text{old})} - \mathbf{Y}_{\cdot j})\} = (\mathbf{H}^{\mathrm{T}}\mathbf{R}\mathbf{H})^{-1}\mathbf{H}^{\mathrm{T}}\mathbf{R}\mathbf{z},
\end{aligned}
\tag{9}
$$

where $\mathbf{R}_{kj} = -\sum_{n=1}^{N}\mathbf{H}_n\mathbf{W}_{\cdot k}^{(\text{old})}(\mathbf{I}_{kj} - \mathbf{H}_n\mathbf{W}_{\cdot j}^{(\text{old})})$ is a weighing matrix and $\mathbf{I}_{kj}$ is the element of the identity matrix, and $\mathbf{z} = \mathbf{H}\mathbf{W}_{\cdot j}^{(\text{old})} - \mathbf{R}^{-1}(\mathbf{Y}_{\cdot j} - \mathbf{W}_{\cdot j}\mathbf{H})$ is an *effective target value*. Eq. (9) takes the form of a set of *normal equations* for a weighted least-squares problem. As the weighing matrix $\mathbf{R}$ is not constant but depends on the parameter vector $\mathbf{W}_{\cdot j}^{(old)}$, we must apply the normal equations iteratively. Each iteration uses the last iteration weight vector $\mathbf{W}_{\cdot j}^{(\text{old})}$ to compute a revised weighing matrix $\mathbf{R}$ and regresses the target value $\mathbf{z}$ with $\mathbf{H}\mathbf{W}_{\cdot j}^{(\text{new})}$. Therefore, the variable decorrelation can also be applied to the GNNs with softmax classifier to reduce the estimation bias in each iteration.

Figure 2(b) is the framework of GNN-DVD, and we input the labeled nodes' embeddings $\tilde{\mathbf{H}}^{(K-1)}$ into the regularizer $\mathcal{L}_{DVD}(\tilde{\mathbf{H}}^{(K-1)})$. As GCN has the formula $softmax(\hat{\mathbf{A}}\mathbf{H}^{(K-1)}\mathbf{W}^{(K-1)})$, the variable weights of $\hat{\mathbf{H}}^{(K-1)}$ used for differentiating $\mathcal{L}_{DVD}(\tilde{\mathbf{H}}^{(K-1)})$ can be computed from $\alpha = \text{Var}(\mathbf{W}^{(K-1)}, \text{axis} = 1)$, where $\text{Var}(\cdot, \text{axis} = 1)$ refers to calculating the variance of each row of some matrix and it reflects each variable's weight for classification which is similar to the regression coefficients. Note that when incorporating VD term with GNNs, we do not need compute the variable weights. Then the sample weights $\mathbf{w}$ learned by DVD term have the ability to remove the correlation in $\tilde{\mathbf{H}}^{(K-1)}$. We propose to use this sample weights to reweight softmax loss:

$$
\min_{\theta} \mathcal{L}_G = \sum_{l \in \mathcal{Y}_L} \mathbf{w}_l \cdot \ln(q(\tilde{\mathbf{H}}_l^{(K)}) \cdot \mathbf{Y}_l),
\tag{10}
$$

where $q(\cdot)$ is the softmax function, $\mathcal{Y}_L$ is the set of labeled node indices and $\theta$ is the set of parameters of GCN. The complexity analysis as well as the optimization of whole algorithm are summarized in Appendix D.

## 4 EXPERIMENTS

**Datasets** Here, we validate the effectiveness of our method on node classification with two kinds of selection biased data, i.e., label selection bias and small sample selection bias. For label selection bias, we empoly three widely used graph datasets: Cora, Citeseer and Pubmed (Sen et al., 2008). As in Section 2.1, we make the inductive setting for each graph and get three biased degrees for each graph. For small sample selection bias, we conduct the experiments on NELL dataset (Carlson

Table 1: Performance of three citation networks. The '*' indicates the best results of the baselines. Best results of all methods are indicated in bold. '% gain over GCN/GAT' means the improvement percent of GCN/GAT-DVD against GCN/GAT, respectively.

| Method | Cora | | | Citeseer | | | Pubmed | | |
|---|---|---|---|---|---|---|---|---|---|
| | Light | Medium | Heavy | Light | Medium | Heavy | Light | Medium | Heavy |
| MLP | 0.5624 | 0.5197 | 0.5087 | 0.4532 | 0.3757 | 0.3893 | 0.6852 | 0.6620 | 0.6378 |
| Planetoid (Yang et al., 2016) | 0.5890 | 0.5240 | 0.5180 | 0.5160 | 0.5140 | 0.4880 | 0.7160 | 0.6770 | 0.6680 |
| Chebyshev (Defferrard et al., 2016) | 0.7116 | 0.7006 | 0.6809 | 0.6542 | 0.6276 | 0.5920 | 0.7358 | 0.6862 | 0.6732 |
| SGC (Wu et al., 2019) | 0.7800 | 0.7800 | 0.7530 | 0.6780 | $0.6730^*$ | 0.6200 | $0.7880^*$ | 0.7560 | 0.6800 |
| APPNP (Klicpera et al., 2019) | 0.7913 | 0.7689 | 0.7629 | 0.6478 | 0.6052 | 0.5903 | 0.7639 | 0.7369 | 0.6862 |
| GNM-GCN (Zhou et al., 2019) | 0.7423 | 0.7531 | 0.7196 | 0.5793 | 0.5717 | 0.5125 | 0.7552 | 0.7381 | 0.7072 |
| GNM-GAT (Zhou et al., 2019) | 0.7875 | 0.7638 | 0.7404 | 0.6524 | 0.6487 | 0.5865 | 0.7438 | 0.7568 | 0.6891 |
| GCN (Kipf & Welling, 2016) | 0.7851 | 0.7775 | 0.7422 | 0.6786 | 0.5952 | 0.5551 | 0.7673 | 0.7545 | 0.7247 |
| GCN-VD | 0.7951 | 0.7855 | 0.7522 | 0.6844 | 0.6676 | 0.6408 | 0.7727 | 0.7729 | 0.7399 |
| GCN-DVD | 0.7959 | 0.7885 | 0.7555 | 0.6908 | 0.6769 | 0.6496 | 0.7741 | **0.7746** | **0.7542** |
| % gain over GCN | 1.38% | 1.41% | 1.79% | 1.8% | 14.2% | 17.0% | 0.89% | 2.67% | 4.07% |
| GAT (Veličković et al., 2017) | $0.8067^*$ | $0.8019^*$ | 0.7578 | $0.7033^*$ | 0.6683 | $0.6475^*$ | 0.7665 | $0.7579^*$ | 0.7068 |
| GAT-VD | 0.8146 | 0.8079 | **0.7708** | 0.7149 | **0.6833** | 0.6611 | 0.7783 | 0.7689 | 0.7149 |
| GAT-DVD | **0.8179** | **0.8119** | 0.7694 | **0.7172** | 0.6825 | **0.6627** | 0.7788 | 0.7723 | 0.7210 |
| % gain over GAT | 1.39% | 1.26% | 1.53% | 1.97% | 2.12% | 2.34% | 1.6% | 1.9% | 2.0% |

et al., 2010) that each class only has one labeled node for training. Due to the large scale of this dataset, the test nodes are easily to have distribution shift from training nodes. The details of the datasets and experimental setup are given in Appendix E. One can download codes and datasets for all experiments from the supplementary material.

**Baselines** Under our proposed framework, we incorporate the VD/DVD term with GCN and GAT called GCN-VD/DVD and GAT-VD/DVD (details in Appendix F), and thus GCN and GAT are two basic baselines. We compare with GNM-GCN/GAT (Zhou et al., 2019) that considers the label selection bias in transductive setting. Moreover, several state-of-the-art GNNs are included: Chebyshev filter (Kipf & Welling, 2016), SGC (Wu et al., 2019) and APPNP (Klicpera et al., 2019). Additionally, we compare with Planetoid (Yang et al., 2016) and MLP trained on the labeled nodes.

**Results on Label Selection Bias Dataset** The results are given in Table 1, and we have the following observations. First, the proposed models (i.e., GCN/GAT with VD/DVD terms) always achieve the best performances in most cases, which well demonstrates that the effectiveness of our proposed debiased GNN framework. Second, comparing with base models, our proposed models all achieve up to 17.0% performance improvements, and gain larger improvements under heavier bias scenarios. Since the major difference between our model with base models is the VD/DVD regularizer, we can safely attribute the significant improvements to the effective decorrelation term and its seamless joint with GNN models. Moreover, GCN/GAT-DVD achieve better results that GCN/GAT-VD in most cases. It validates the importance and effectiveness of differentiating variables' weights in semi-supervised setting. Additional experimental results about the sample weight analysis and parameter sensitivity analysis can be found in Appendix G.

**Results on Small Sample Selection Bias Dataset** As NELL is a large-scale graph, we cannot run GAT on a single GPU with 16GB memory. We only perform GCN-VD/DVD and compare with representative methods which can perform on this dataset. The results are shown in Table 2. First, GCN-VD/DVD achieve significant improvements over GCN. It indicates that selection bias could be induced by a small number of labeled nodes and our proposed method can relieve the estimation bias. Moreover, GCN-DVD further improves GCN-VD with a large margin. It further validates that decorrelating all the variable pairs equally is suboptimal, and our differentiated strategy is effective when labeled nodes are scarce. The reason that GNM-GCN fails is the GNM relies on the accuracy of the IPW estimator that predicts the probability of a node to be selected, however, in this dataset, the ratio of positive and negative samples are extremely unbalanced influencing the performance of IPW.

Table 2: Performance of NELL

| Dataset | MLP | Planetoid | SGC | GNM-GCN | GCN | GCN-VD | GCN-DVD |
|---|---|---|---|---|---|---|---|
| NELL | 0.2385 | 0.3901 | 0.4128 | 0.1589 | 0.4416 | 0.4652 | **0.4734** |

## 5 RELATED WORKS

In the past few years, Graph Neural Networks (GNNs) (Scarselli et al., 2008; Kipf & Welling, 2016; Veličković et al., 2017; Xu et al., 2019; Klicpera et al., 2019) have become the major technology to capture patterns encoded in the graph due to its powerful representation capacity. Although the current GNNs have achieved great success, when applied to inductive setting, they all assume that training nodes and test nodes follow the same distribution. However, this assumption does not always hold in real applications. GNM (Zhou et al., 2019) first pays attention on the label selection problem on graph learning, and it learns a IPW estimator to estimate the probability of each node to be selected and uses this probability to reweight the labeled nodes. However, it heavily relies on the accuracy of IPW estimator, which depends on the label assignment distribution of whole graph, hence it is more suitable for transductive setting.

To enhance the stability in unseen varied distributions, some literatures (Shen et al., 2020b; Kuang et al., 2020) have revealed the connection between correlation and prediction stability under model misspecification. However, these methods are built on the simple regressions, but GNNs have more complex structure and properties needed to be considered. We also notice that Shen et al. (2020a) propose a differentiated variable decorrelation term for linear regression. However, this decorrelation term requires multiple environment with different correlations between stable variable and unstable variable available in the training stage while our method do not require.

## 6 CONCLUSION

In this paper, we investigate a general and practical problem: learning GNNs with agnostic selection bias. The selection bias will inevitably cause the GNNs to learn the biased correlation between aggregation mode and class label and make the prediction unstable. We then propose a novel differentiated decorrelated GNN, which combines the debiasing technique with GNNs in a unified framework. Extensive experiments well demonstrate the effectiveness and flexibility of GNN-DVD.

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

## A   DERIVATION OF $\widehat{MTEF}$

$$
\begin{aligned}
\widehat{MTEF} &= \frac{\sum_{i:T_i=t} \mathbf{w}_i \cdot Y_i(t) - \sum_{j:T_j=t-\Delta t} \mathbf{w}_j \cdot Y_j(t-\Delta t)}{\Delta t} \\
&= \frac{\sum_{i:T_i=t} \mathbf{w}_i \cdot (\sum_{k \neq t} \alpha_k \mathbf{X}_{ik} + \alpha_t t + c + \epsilon) - \sum_{j:T_j=t-\Delta t} \mathbf{w}_j \cdot (\sum_{k \neq t} \alpha_k \mathbf{X}_{jk} + \alpha_t (t-\Delta t) + c + \epsilon)}{\Delta t} \\
&= \frac{\sum_{i:T_i=t} \mathbf{w}_i \alpha_t t - \sum_{j:T_j=t-\Delta t} \mathbf{w}_j \alpha_t (t-\Delta t)}{\Delta t} \\
&\quad + \frac{(\sum_{i:T_i=t} \mathbf{w}_i \sum_{k \neq t} \alpha_k \mathbf{X}_{ik} - \sum_{j:T_j=t-\Delta t} \mathbf{w}_j \sum_{k \neq t} \alpha_k \mathbf{X}_{ik})}{\Delta t} + \phi(\epsilon) \\
&= MTEF + \sum_{k \neq t} \alpha_k (\frac{\sum_{i:T_i=t} \mathbf{w}_i \cdot \mathbf{X}_{ik} - \sum_{j:T_j=t-\Delta t} \mathbf{w}_j \cdot \mathbf{X}_{jk}}{\Delta t}) + \phi(\epsilon),
\end{aligned}
\tag{11}
$$

where $\frac{\sum_{i:T_i=t} \mathbf{w}_i \alpha_t t - \sum_{j:T_j=t-\Delta t} \mathbf{w}_j \alpha_t (t-\Delta t)}{\Delta t}$ is the ground truth of $MTEF$, $\phi(\epsilon)$ means the noise term, and $\phi(\epsilon) \simeq 0$ with Gaussian noise.

## B   PROOF OF THEOREM 2

$$
\hat{\mathbf{w}} = \arg\min_{\mathbf{w}} \sum_{j=1}^{p} (\alpha^{\mathrm{T}} \cdot \mathrm{abs}(\mathbf{H}_{.j}^{\mathrm{T}} \Lambda_{\mathbf{w}} \mathbf{H}_{.-j}/n - \mathbf{H}_{.j}^{\mathrm{T}} \mathbf{w}/n \cdot \mathbf{H}_{.-j}^{\mathrm{T}} \mathbf{w}/n))^2 + \frac{\lambda_1}{n} \sum_{i=1}^{n} \mathbf{w}_i^2 + \lambda_2 (\frac{1}{n} \sum_{i=1}^{n} \mathbf{w}_i - 1)^2
\tag{12}
$$

**Proof**   For simplicity, we denote $\mathcal{L}_1 = \sum_{j=1}^{p} (\alpha^{\mathrm{T}} \cdot \mathrm{abs}(\mathbf{H}_{.j}^{\mathrm{T}} \Lambda_{\mathbf{w}} \mathbf{H}_{.-j}/n - \mathbf{H}_{.j}^{\mathrm{T}} \mathbf{w}/n \cdot \mathbf{H}_{.-j}^{\mathrm{T}} \mathbf{w}/n))^2$, $\mathcal{L}_2 = \frac{1}{n} \sum_{i=1}^{n} \mathbf{w}_i^2$, $\mathcal{L}_3 = (\frac{1}{n} \sum_{i=1}^{n} \mathbf{w}_i - 1)^2$ and $\mathcal{F}(\mathbf{w}) = \mathcal{L}_1 + \lambda_1 \mathcal{L}_1 + \lambda_2 \mathcal{L}_2$. We first calculate the Hessian matrix of $\mathcal{F}(\mathbf{w})$, denoted as $\mathbf{H}_e$, to prove the uniqueness of the optimal solution $\hat{\mathbf{w}}$, as follows:

$$
\mathbf{H}_e = \frac{\partial^2 \mathcal{L}_1}{\partial \mathbf{w}^2} + \lambda_1 \frac{\partial^2 \mathcal{L}_2}{\partial \mathbf{w}^2} + \lambda_2 \frac{\partial^2 \mathcal{L}_3}{\partial \mathbf{w}^2}
$$

For the term $\mathcal{L}_1$, we can rewrite it as:

$$
\begin{aligned}
\mathcal{L}_1 &= \sum_{j \neq k} \alpha_i^2 \alpha_k^2 (\frac{1}{n} \sum_{i=1}^{n} \mathbf{H}_{i,j} \mathbf{H}_{i,k} \mathbf{w}_i - (\frac{1}{n} \sum_{i=1}^{n} \mathbf{H}_{i,j} \mathbf{w}_i)(\frac{1}{n} \sum_{i=1}^{n} \mathbf{H}_{i,k} \mathbf{w}_i))^2 \\
&= \sum_{j \neq k} \alpha_i^2 \alpha_k^2 ((\frac{1}{n} \sum_{i=1}^{n} \mathbf{H}_{i,j} \mathbf{H}_{i,k} \mathbf{w}_i)^2 - (\frac{2}{n} \sum_{i=1}^{n} \mathbf{H}_{i,j} \mathbf{H}_{i,k} \mathbf{w}_i)(\frac{1}{n} \sum_{i=1}^{n} \mathbf{H}_{i,j} \mathbf{w}_i)(\frac{1}{n} \sum_{i=1}^{n} \mathbf{H}_{i,k} \mathbf{w}_i) \\
&\quad + ((\frac{1}{n} \sum_{i=1}^{n} \mathbf{H}_{i,j} \mathbf{w}_i)(\frac{1}{n} \sum_{i=1}^{n} \mathbf{H}_{i,k} \mathbf{w}_i))^2)
\end{aligned}
$$

And when $|\mathbf{H}_{i,j}| \leq c$, for any variable $j$ and $k$, and $|\mathbf{w}_i| \leq c$, we have $\frac{\partial^2}{\partial \mathbf{w}^2} (\frac{1}{n} \sum_{i=1}^{n} \mathbf{H}_{i,j} \mathbf{H}_{i,k} \mathbf{w}_i)^2 = \mathcal{O}(\frac{1}{n^2})$, $\frac{\partial^2}{\partial \mathbf{w}^2} (\frac{1}{n} \sum_{i=1}^{n} \mathbf{H}_{i,j} \mathbf{w}_i)(\frac{1}{n} \sum_{i=1}^{n} \mathbf{H}_{i,k} \mathbf{w}_i) = \mathcal{O}(\frac{1}{n^2})$ and $\frac{\partial^2}{\partial \mathbf{w}^2} ((\frac{2}{n} \sum_{i=1}^{n} \mathbf{H}_{i,j} \mathbf{H}_{i,k} \mathbf{w}_i)(\frac{1}{n} \sum_{i=1}^{n} \mathbf{H}_{i,j} \mathbf{w}_i)(\frac{1}{n} \sum_{i=1}^{n} \mathbf{H}_{i,k} \mathbf{w}_i)) = \mathcal{O}(\frac{1}{n^2})$. Then with $|\alpha_i| \leq c$, we have $\alpha_i^2 \alpha_k^2 \frac{\partial^2}{\partial \mathbf{w}^2} (\frac{1}{n} \sum_{i=1}^{n} \mathbf{H}_{i,j} \mathbf{H}_{i,k} \mathbf{w}_i - (\frac{1}{n} \sum_{i=1}^{n} \mathbf{H}_{i,j} \mathbf{w}_i)(\frac{1}{n} \sum_{i=1}^{n} \mathbf{H}_{i,k} \mathbf{w}_i))^2 = \mathcal{O}(\frac{1}{n^2})$. $\mathcal{L}_1$ is sum of $p(p-1)$ such terms. Then we have

$$
\frac{\partial^2 \mathcal{L}_1}{\partial \mathbf{w}^2} = \mathcal{O}(\frac{p^2}{n^2}).
$$

With some algebras, we can also have

$$
\frac{\partial^2 \mathcal{L}_2}{\partial \mathbf{w}^2} = \frac{1}{n} \mathbf{I},
$$

$$\frac{\partial^2 \mathcal{L}_3}{\partial \mathbf{w}^2} = \frac{1}{n^2} \mathbf{1} \mathbf{1}^{\mathrm{T}},$$

thus,

$$\mathbf{H}_e = \mathcal{O}(\frac{p^2}{n^2}) + \frac{\lambda_1}{n} \mathbf{I} + \frac{\lambda_2}{n^2} \mathbf{1} \mathbf{1}^{\mathrm{T}} = \frac{\lambda_1}{n} \mathbf{I} + \mathcal{O}(\frac{p^2 + \lambda_2}{n^2}).$$

Therefore, if $\frac{\lambda_1}{n} \gg \frac{p^2 + \lambda_2}{n^2}$, equivalent to $\lambda_1 n \gg p^2 + \lambda_2$, $\mathbf{H}_e$ is an almost diagonal matrix. Hence, $\mathbf{H}_e$ is positive definite (Nakatsukasa, 2010). Then the function $\mathcal{F}(\mathbf{w})$ is convex on $\mathcal{C} = \{\mathbf{w} : |\mathbf{w}_i| \le c\}$, and has unique optimal solution $\hat{\mathbf{w}}$.

Moreover, because $\mathcal{L}_1$ is our major decorrelation term, we hope $\mathcal{L}_1$ to dominate the terms $\lambda_1 \mathcal{L}_2$ and $\lambda_2 \mathcal{L}_3$. On $\mathcal{C}$, we have $\mathcal{L}_1 = \mathcal{O}(1)$, $\mathcal{L}_2 = \mathcal{O}(1)$, and $\alpha_i^2 \alpha_k^2 (\frac{1}{n} \sum_{i=1}^n \mathbf{H}_{i,j} \mathbf{H}_{i,k} \mathbf{w}_i - (\frac{1}{n} \sum_{i=1}^n \mathbf{H}_{i,j} \mathbf{w}_i)(\frac{1}{n} \sum_{i=1}^n \mathbf{H}_{i,k} \mathbf{w}_i))^2 = \mathcal{O}(1)$. Thus $\mathcal{L}_1 = \mathcal{O}(p^2)$. When $p^2 \gg \max(\lambda_1, \lambda_2)$, $\mathcal{L}_1$ will dominate the regularization terms $\mathcal{L}_2$ and $\mathcal{L}_3$.

## C   PROOF OF THEOREM 3

Let $\mathbf{Z} = \{\mathbf{Z}_1, \mathbf{Z}_2, \cdots, \mathbf{Z}_p\}$ be $p$ pairwise uncorrelated variables. $\forall \mathbf{Z}_i, \mathbf{Z}_j \in \mathbf{Z}$, $(\mathbf{Z}_i^{(1)}, \mathbf{Z}_i^{(2)}, \cdots, \mathbf{Z}_i^{(n)})$ and $(\mathbf{Z}_j^{(1)}, \mathbf{Z}_j^{(2)}, \cdots, \mathbf{Z}_j^{(n)})$ are $n$ *simple random samples* drawn from $\mathbf{Z}_i$ and $\mathbf{Z}_j$ respectively, and have same distribution with $\mathbf{Z}_i$ and $\mathbf{Z}_j$. Given a linear aggregation matrix $\hat{\mathbf{A}} = (a_{ij})$, $\forall s, v \in (1, 2, \cdots, n)$, let $\mathbf{Y}_i^{(s)} = \sum_{k=1}^n a_{sk} \mathbf{Z}_i^{(k)}$ and $\mathbf{Y}_j^{(v)} = \sum_{l=1}^n a_{vl} \mathbf{Z}_j^{(l)}$, and we have following derivation:

$$\mathrm{Cov}(\mathbf{Y}_i^{(s)}, \mathbf{Y}_j^{(v)}) = \mathrm{Cov}(\sum_{k=1}^n a_{sk} \mathbf{Z}_i^{(k)}, \sum_{l=1}^n a_{vl} \mathbf{Z}_j^{(l)})$$

$$= \sum_{k=1}^n \sum_{l=1}^n a_{sk} a_{vl} \mathrm{Cov}(\mathbf{Z}_i^{(k)}, \mathbf{Z}_j^{(l)}) = \sum_{k=1}^n \sum_{l=1}^n a_{sk} a_{vl} \delta_{ij},$$

where $\delta_{ij} = 0$ when $i \ne j$, otherwise $\delta_{ij} = 1$. Therefore, when $i \ne j$, we have $\mathrm{Cov}(\mathbf{Y}_i^{(s)}, \mathbf{Y}_j^{(v)}) = 0$ and $\mathrm{Cov}(\mathbf{Y}_i, \mathbf{Y}_j) = 0$. Extended the conclusion to multiple variable, $\mathbf{Y} = (\mathbf{Y}_1, \mathbf{Y}_2, \cdots, \mathbf{Y}_n)$ are pairwise uncorrelated. Proof completes.

## D   PSEUDOCODE OF GNN-DVD

---

**Algorithm 1:** GNN-DVD Algorithm

---

| **Input** | : Training graph $\mathcal{G}_{train} = \{\mathbf{A}, \mathbf{X}, \mathbf{Y}\}$, and indices of labeled nodes $\mathcal{Y}_L$; Max iteration: $maxIter$ |
| **Output** | : GNN parameter $\theta$ and sample weights $\mathbf{w}$ |
| **Initialization** | : Let $\mathbf{w} = \omega \odot \omega$ and initialize sample weights $\omega$ with $\mathbf{1}$; Initialize GNN's parameters $\theta$ with random uniform distribution; Iteration $t \leftarrow 0$ |

1  **while** *not converged* or $t < maxIter$ **do**
2      Optimize $\theta^{(t)}$ to minimize $\mathcal{L}_G$;
3      Calculate variable weights $\alpha^{(t)}$ from $\mathbf{W}^{(K-1)}$;
4      Optimize $\omega^{(t)}$ to minimize $\mathcal{L}_{DVD}(\tilde{\mathbf{H}}^{(K-1)})$;
5      $t = t + 1$;
6  **end**
7  **Return:** $\theta$ and $\mathbf{w} = \omega \odot \omega$

---

To optimize our GNN-DVD algorithm, we propose an iterative method. Firstly, we let $\mathbf{w} = \omega \odot \omega$ to ensure non-negativity of $\mathbf{w}$ and initialize sample weight $\omega_i = 1$ for each sample $i$ and GNN's parameters $\theta$ with random uniform distribution. Once the initial values are given, in each iteration,

we fix the sample weights $\omega$ and update the GNN's parameters $\theta$ by $\mathcal{L}_G$ with gradient descent, then compute the confounder weights $\alpha$ from the linear transform matrix $\mathbf{W}^{(K-1)}$. With $\alpha$ and fixing the GNN's parameters $\theta$, we update the sample weights $\omega$ with gradient descent to minimize $\mathcal{L}_{DVD}(\mathbf{H}^{(K-1)})$. We iteratively update the sample weights $\mathbf{w}$ and GNN's parameters $\theta$ until $\mathcal{L}_G$ converges.

**Complexity Analysis** Compared with base model (e.g., GCN and GAT), the mainly incremental time cost is the complexity from DVD term. The complexity of DVD term is $\mathcal{O}(np^2)$, where $n$ is the number of labeled nodes and $p$ is the dimension of embedding. And it is quite smaller than the base model (e.g., the complexity of GCN is linear to the number of edges).

## E    DATASET DESCRIPTION AND EXPERIMENTAL SETUP

### E.1    DATASET DESCRIPTION

Table 3: Dataset statistics

| Dataset | Type | Nodes | Edges | Classes | Features | Bias degree ($\epsilon$) | Bias type |
|---------|------|-------|-------|---------|----------|--------------------------|-----------|
| Cora | Citation network | 2,708 | 5,429 | 7 | 1,433 | 0.7/0.8/0.9 | Label selection bias |
| Citeseer | Citation network | 3,327 | 4,732 | 6 | 3,703 | 0.7/0.8/0.9 | Label selection bias |
| Pubmed | Citation network | 19,717 | 44,338 | 3 | 500 | 0.7/0.8/0.9 | Label selection bias |
| NELL | Knowledge graph | 65,755 | 266,144 | 210 | 5,414 | One labeled node per class | Small sample selection bias |

Some statistics of datasets used in our paper are presented in Table 3, including the number of nodes, the number of edges, the number of classes, the number of features, the bias degree $\epsilon$ and bias type. For three citation networks, we conduct the biased labeled node selection process to get three degrees of datasets for each dataset to validate the effect of label selection bias, in which each class in each dataset contains 20 labeled nodes in training set and the validation set and test set are same as Yang et al. (2016). For NELL, because it only has a single labeled node per class in training set, the training nodes are hard to cover all the neighborhood distribution happened in the test set. Hence, we use this dataset to validate the effectiveness of our method on the extreme small labeled nodes size bias. The data splits are also same as Yang et al. (2016). A description of each of dataset is given as follows:

- Cora (Sen et al., 2008) is a citation network of Machine Learning papers that collected from 7 classes:{Theory, Case Based, Reinforcement Learning, Genetic Algorithms, Neural Networks, Probabilistic Methods, Rule Learning }. Nodes represent papers, edges refer to the citation relationship, and features are bag-of-words vectors for each paper.

- Citeseer (Sen et al., 2008) is a citation network of Machine Learning papers that collected from 6 classes:{Agents, Artificial Intelligence, Database, Information Retrieval, Machine Learning, Human Computer Interaction }. Nodes represent papers, edges refer to the citation relationship, and features are bag-of-words vectors for each paper.

- Pubmed (Sen et al., 2008) is a citation network from the PubMed database, which contains a set of articles (Nodes) related to diabetes and the citation relationship among them. The node features are bag-of-words vectors, and the node label are the diabetes type researched in the articles.

- NELL Carlson et al. (2010) is a dataset extracted from the knowledge graph, which is a set of entities connected with directed, labeled edges (relations). Our pre-processing scheme is same as Yang et al. (2016), where each entity pair $(e_1, r, e_2)$ is assigned with separate relation nodes $r_1$ and $r_2$ as $(e_1, r_1)$ and $(e_2, r_2)$. We use text bag-of-words representation as feature vector of the entities.

### E.2    EXPERIMENTAL SETUP

As the Section 2.1 has described, for all datasets, to simulate the agnostic selection bias scenario, we first follow the inductive setting in Wu et al. (2019) that masks the validation and test nodes in the training phase and validation and test with whole graph so that the test nodes will be agnostic. For GCN and GAT, we utilize the same two-layer architecture as their original paper (Kipf & Welling, 2016; Veličković et al., 2017). We use the following sets of hyperparameters for GCN on Cora,

Citeseer, Pubmed: 0.5 (dropout rate), $5 \cdot 10^{-4}$ (L2 regularization) and 32 (numbder of hidden units); and for NELL: 0.1 (dropout rate), $1 \cdot 10^{-5}$ (L2 regularization) and 64 (number of hidden units). For GAT on Cora, Citeseer, we use: 8 (first layer attention heads), 8 (features each head), 1 (second layer attention head), 0.6 (dropout), 0.0005 (L2 regularization); and for Pubmed: 8 (second layer attention head), 0.001 (L2 regularization), other parameters are same with Cora and Citeseer. To fair comparison, the GNN part of our model uses the same architecture and hyper-parameters with base model and we grid search $\lambda_1$ and $\lambda_2$ from $\{0.01, 0.1, 1, 10, 100\}$. For other baselines, we use the optimal hyper-parameters in literatures on each dataset. For all the experiments, we run 10 times with different random seed and report its average Accuracy results.

## F   EXTEND TO GAT

We can easily incorporate VD/DVD term to other GNNs. We combine them with GAT and more extensions leave as future work. GAT utilizes attention mechanism to aggregate neighbor information. It also follows the linear aggregation and transformation steps. Similar with GCN, the hidden embedding $\tilde{\mathbf{H}}^{(K-1)}$ is the input of VD/DVD term, and the variable weights $\alpha$ are calculated from the transformation matrix $\mathbf{W}^{(K-1)}$ and the sample weights $\mathbf{w}$ are used to reweight the softmax loss. Note that original paper utilizes same transformation matrix $\mathbf{W}^{(K-1)}$ for transforming embedding and learning attention values. Because $\alpha$ means the importance of each variable for classification, and it should be computed from transformation matrix $\mathbf{W}^{(K-1)}$ for transforming embedding, hence we use separate matrix for transforming embedding and learning attention values respectively. This modification does not change the performance of GAT in experiments.

## G   ADDITIONAL EXPERIMENTS

### G.1   SAMPLE WEIGHT ANALYSIS

Here we analyze the effect of sample weights $\mathbf{w}$ in our model. We compute the amount of correlation in the labeled nodes' embeddings $\tilde{\mathbf{H}}^{(K-1)}$ learned by standard GCN and the weighted embeddings of the same layer learned by GCN-DVD. Note that, the weights are the last iteration of sample weights of GCN-DVD. Following Cogswell et al. (2016), the amount of correlation of GCN and GCN-DVD are measured by Frobenius norm of cross-corvairance matrix computed from vectors of $\tilde{\mathbf{H}}^{(K-1)}$ and weighted $\tilde{\mathbf{H}}^{(K-1)}$ respectively. Figure 3 shows the amount of correlation in unweighted and weight embeddings, and we can observe that the embeddings' correlation in all datasets are reduced, demonstrating that the weights learned by GCN-DVD can reduce the correlations between embedded variables. Moreover, one can observe that it is hard to reduce the correlation to zero. Therefore, the necessity of differentiating variables' weights will be further validated.

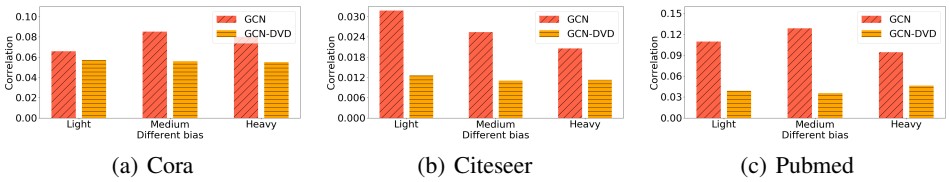

| (a) Cora | (b) Citeseer | (c) Pubmed |

Figure 3: Embedding correlation analysis on unweighted and weighted GCN.

### G.2   PARAMETER SENSITIVITY

We study the sensitiveness of parameters and report the results of GCN-DVD on three citation networks in Fig. 4-6. The experimental results show that GCN-DVD is relatively stable to $\lambda_1$ and $\lambda_2$ with wide ranges in most cases, indicating the robustness of our model.

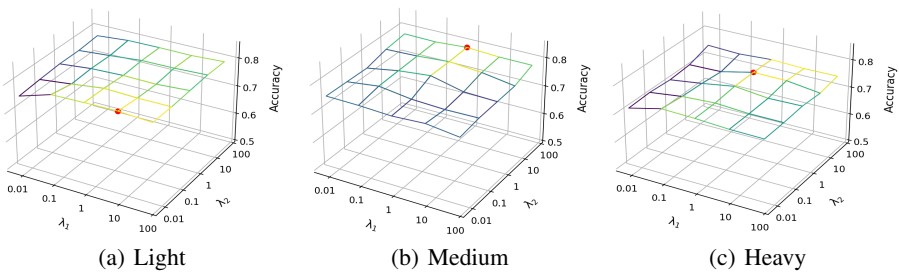

(a) Light        (b) Medium        (c) Heavy

Figure 4: Accuracy of GCN-DVD with different $\lambda_1$ and $\lambda_2$ on different biased Cora datasets.

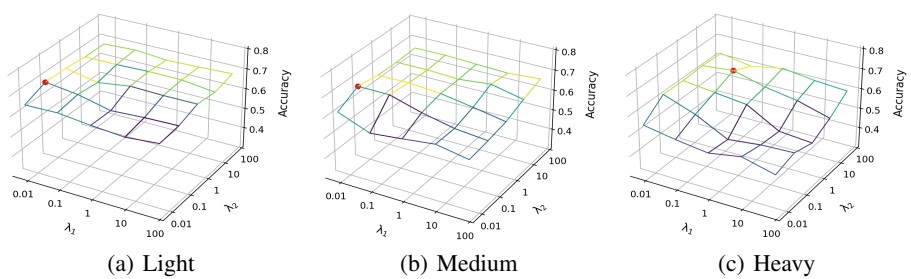

(a) Light        (b) Medium        (c) Heavy

Figure 5: Accuracy of GCN-DVD with different $\lambda_1$ and $\lambda_2$ on different biased Citeseer datasets.

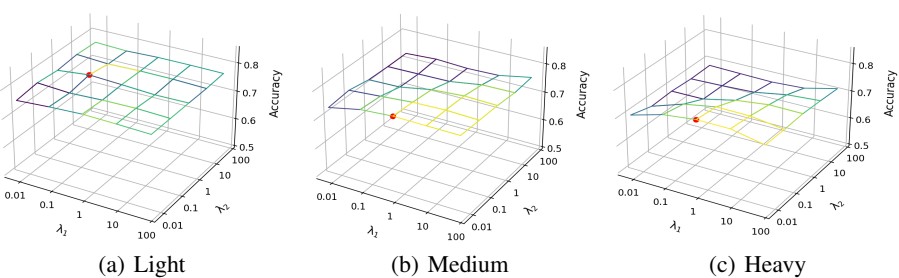

(a) Light        (b) Medium        (c) Heavy

Figure 6: Accuracy of GCN-DVD with different $\lambda_1$ and $\lambda_2$ on different biased Pubmed datasets.

