# OpenReview forum: "Debiased Graph Neural Networks with Agnostic Label Selection Bias"
_ICLR.cc/2021/Conference — Reject_

### Official Review · AnonReviewer4 · 2020-10-25
**An innovative work for an interesting problem**

**Rating:** 8
**Confidence:** 4

**Review:**

The paper proposes an important and unexplored problem in GNNs, i.e., the inconsistent distribution between the training set with test set caused by agnostic label selection bias. I believe that studying this problem is very important for generalizing GNNs on unseen test nodes. The paper first conducts an investigated experiment to show the great impact of agnostic selection bias on test performance. Moreover, the theoretical analysis is provided to identify how the label selection bias leads to the estimation bias in GNN parameters.
To remove the estimation bias in parameter estimation, the paper proposes a novel DGNN framework by jointly optimizing a differentiated decorrelation regularizer (DVD) and a weighted GNNs model. The DVD regularizer is designed based on the causal view of variable decorrelation terms. I personally like the idea of analyzing variable decorrelation by the casual view. Furthermore, the paper theoretically proves that how to combine variable decorrelation terms with GNNs would be a more flexible framework for most GNNs and how to extend the theory to the multi-classification scenario. Overall, the proposed method is theoretical sound, where some basic claims are all supported by the clear and sound theoretical analysis.
The paper conducts extensive experiments on four benchmark datasets with two kinds of selection bias, well showing the effectiveness of the proposed model. Basically, the paper is well motivated and well-organized.

Strong points:
1.	The agnostic label selection bias problem in GNNs proposed by this paper is very important but seldom studied. And the paper shows the effect of label selection bias on the generalization of GNN in both experimental and theoretical way. In practice, the selection bias widely exists, I think this work may attract more attention in this direction, which makes GNNs more robust and stable in unseen environments.
2.	The technique of the proposed method is sound. The differentiated variable decorrelation is well motivated. This is a general framework for enhancing most existing GNNs under label selection bias setting. The idea of analysis and design model is novel, for example, analyze the estimation bias with stable learning theory, differentiated variable decorrelation in causal view, prove how to combine DVD with GNNs is more flexible, and extend the method to the multi-classification setting. I think these ideas are instructive.
3.	The experiment part is comprehensive and convincing. The experiments are conducted on two kinds of selection bias data, i.e., label selection bias and small sample selection bias. These two kinds of selection bias usually happen in real-world scenarios. And the results clearly show that the proposed methods make larger improvements with heavier bias.

Question for rebuttal:
1.	In section 3.3, the variable weight \alpha is computed from Var(W^(K−1), axis = 1), and \alpha_i can only be a positive value, however, in linear regression, the coefficients could also be a negative value. Hence, how to keep the \alpha computed from Var(W^(K−1), axis = 1) has the same meaning of linear regression coefficients?
2.	Although we can find the hyperparameters for each method from the experiment part and the corresponding paper of baselines, it is better to list all the hyperparameters used in the paper in the Appendix to improve reproducibility.

---

> ### Author Response · Authors · 2020-11-19
> **Response to Reviewer #4(Part 1/1)**
>
> Thanks for the reviewer's positive feedback. The reviewer summarizes the strong points properly and also points out valuable suggestions to improve the paper. We respond the reviewer's concerns as follows.
>
> Q1. How to keep the $\alpha$ computed from Var($\mathbf{W}^{(K−1)}$, axis = 1) has the same meaning of linear regression coefficients?
>
> Given the transformation matrix $\mathbf{W}^{(K-1)}\in\mathbb{R}^{p\times K}$ and a node embedding $\mathbf{H}_\textrm{l}\in\mathbb{R}^{p\times 1}$,  if we want to classify node embedding into one of $K$ classes by $\mathbf{W}^{(K-1)}$ and softmax layer, $\mathbf{H}_\textrm{l}^{\text{T}}\cdot\mathbf{W}^{(K-1)}_\textrm{.j}$ will indicate the similarity between node $l$ and class $j$.  Hence,  $\mathbf{W}^{(K-1)}_\textrm{.j}\in\mathbb{R}^{p\times 1}$ can be viewed as the class center for $j$-th class and the variance of each row of $\mathbf{W}^{(K-1)}_\textrm{.j}$ will reflect each variable's weight for classification which is similar to linear regression coefficients.
>
> Q2. It is better to list all the hyperparameters used in the paper in the Appendix to improve reproducibility.
>
> Thank you for your kindly suggestion, and it helps to improve our work's reproducibility. We have added all necessary hyperparameters in the Appendix to reproduce the results in the paper.
>
> Thanks again for your useful and positive feedback about our work. If you have any other concerns, please let us know.

---

### Official Review · AnonReviewer3 · 2020-10-28

**Rating:** 4
**Confidence:** 2

**Review:**

This paper presents a novel method to remove the selection bias of graph data, which is neglected by previous methods. Specifically, the authors suspect that all variables observed by GNNs can be decomposed into two parts, stable variables and unstable variables.  Then, DGNN, a differentiable decorrelation regularization is proposed to reweight each variable pair to eliminate estimation bias. Experiments on three datasets confirm its effectiveness.

Pros:
+ The studied problem is general and also practical for real-world applications.

Cons:
+ The novelty of this work is limited. Although the authors claim it is the first work to solve agnostic label selection bias problem, I in person believe this work can be regarded as a special case of DWR [1]. Therefore, on the basis of DWR, this paper presents not much theoretical contribution to this problem.
+ The presentation of this paper is somewhat confusing and not well-motivated. For example, it is not clear to understand the connection between the example presented in Section 2.1 and the proposed method. Also, why does this paper consider the Newton-Raphson update rule in Equation (9)? Besides, how do you efficiently compute the inversion of matrices?
+ The studied datasets are known to have unstable performance and are also of small scales. Even so, the performance improvement seems to be marginal with new baselines missing. Larger datasets such as OGB are strongly encouraged.

Reference:
[1] Stable Prediction with Model Misspecification and Agnostic Distribution Shift, AAAI 2020.

---

> ### Author Response · Authors · 2020-11-19
> **Response to Reviewer #3(Part 1/2)**
>
> We thank the reviewer for her/his insightful comments and discuss the concerns raised point by point below.
>
> Q1.  The novelty of this work is similar to DWR.
>
> Our work is actually quite different from DWR, which can be summarized as follows:
>
> - In the problem level, DWR focuses on attributed data, while our work studies a new important problem in graph learning, i.e., graph learning with agnostic label selection bias. This problem has not been explored by previous literatures. For this new problem, we clarify several issues, which are unclear before. For example, the distribution shift may be induced by label selection bias and small number of selected nodes，and the selection bias would drastically hinder the generalization ability of GNNs.
> - In the theoretical level, we make the following contributions: (1) We analyze the relationship between the raw input and the learned embedding, and how the spurious correlation of stable variables and unstable variables will induce the parameter estimation bias. (Section 2.2) (2) We bridge the gap between treatment effect estimation with variable decorrelation and apply the differentiated theorem proposed by this work to reweight the variable weights in the variable decorrelation term. (Section 3.1 and 3.2.) (3) We theoretically prove how to make our framework feasible to popular GNNs and classification task. (Section 3.3.)
> - In the model level, DWR is built  on simple regression, which cannot be directly applied to graph data. And GNNs have a more complex structure and properties needed to be considered. Different from the variable decorrelation term proposed in DWR, our work proposes a differentiated variable decorrelation term that is more suitable for semi-supervised learning. Moreover, we prove how to incorporate VD/DVD with GNNs is a more flexible framework,  which can easily incorporate with various popular GNNs.
>
> Q2.  It is not  clear to understand the connection between the example presented in Section 2.1 and the proposed method.
>
> The example presented in Section 2.1 is used to better demonstrate that the problem we studied does exist in real graphs, i.e., the state-of-the-art GNNs are sensitive to the selection bias. Based on the proposed problem, we provide detailed analysis and propose new method.  Particularly, we perform GNNs on three biased graph datasets. And the results show that the selection bias drastically hinders the generalization ability on test nodes. Based on this experimental investigation, we theoretically analyze the label selection bias will affect the parameter estimation bias on GNNs in Section 2.2. And the parameter estimation bias is mainly induced by the  spurious correlation between stable variables $\mathbf{S}$ and unstable variables $\mathbf{V}$. If we can remove the bias term in parameter estimation, we can achieve the stable prediction. To achieve this goal, we propose to decorrelate the embedding of GNNs so that the spurious correlation could be removed.
>
> In conclusion, the experimental investigation in Section 2.1 motivates our problem, and theoretical analysis in Section 2.2 bridges the gap between data selection bias with parameter estimation bias. Finally, our model is proposed based on the theoretical analysis results of Eq.(3) and Eq.(4) in Section 2.2. Therefore, our method has a close connection with the example in Section 2.1.
>
> Q3.  Why does this paper consider the Newton-Raphson update rule in Equation (9)? How do you efficiently compute the inversion of matrices?
>
> In Section 2.2, we analyze GNNs based on the least-squares loss. However, most GNNs are designed for the classification task with the cross-entropy loss. In order to be theoretically rigorous, we extend our theory from regression task to multi-classification task. The intention of the Newton-Raphson update rule presented in Eq. (9) is to indicate that every update iteration of GNN with softmax classifier can be viewed as a weighted least-squares problem. Hence, the theory derived from the least-squares loss can also be applied to classification task. The Newton-Raphson update rule is not the update method used in the paper, and it is the thoery to bridge the gap between linear regression and multi-classification. In practice, we use Adam optimizer to optimize the method,  so we do not need to compute the inversion of matrices.

---

> > ### Author Response · Authors · 2020-11-19
> > **Response to Reviewer #3(Part 2/2)**
> >
> > Q4. The studied datasets are known to have unstable performance and are also of small scales. Larger datasets such as OGB are strongly encouraged.
> >
> > Thank you for your valuable suggestion. We conduct our experiments on a total of four datasets. In Table 1, we use three widely used datasets in graph learning literatures. Different from the original datasets, we modify each dataset with three different degrees of training label selection bias. And from the results, we know that our methods consistently outperform SOTA baselines and gain larger improvements under heavier bias scenarios. Moreover, in Table 2, we conduct our experiments on a large-scale dataset, NELL,  which has 65,755 nodes, 266,144 edges and only one labeled node for each class node for training, to validate the effectiveness of our method on the small number of labeled node selection bias. For more details about our datasets, please refer to Appendix F.1. It is promising to validate our method on more datasets, but limited by the time of rebuttal we have to leave it as future work.
> >
> > Thanks for the reviewer's all valuable comments. We will polish the paper in the revision, considering your suggestions.  If you have any other concerns, please let us know.

---

### Official Review · AnonReviewer2 · 2020-10-28
**Marginal improvement and inconclusive evaluation**

**Rating:** 5
**Confidence:** 4

**Review:**

Summary:
The authors propose two different regularization terms to help mitigate the effect of label selection bias. The regularizers are well motivated and can be applied to different GNN models.

Reasons for score:
Overall, I recommend a weak reject. While the theoretical analysis is interesting, the experimental evaluation is inconclusive and the performance improvement is marginal (see weak points). If the authors show stronger empirical evidence (see questions) I will consider increasing the score. Moreover, it is not clear whether the type of selection bias studied in the paper is actually relevant in practice.

Strong points:
* The proposed regularizers are well motivated, theoretically supported and at the same time simple and easy to implement.
* The causal view analysis of the proposed regularizers is insightful.
* The regularized have a reasonably small computational complexity.

Weak points:
* There are no results in the paper which show how the proposed method performs for a standard (non biased) labeling scenario. It is not clear whether the performance of GCN/GAT-VD/DVD in the standard setting is worse, roughly the same or better, and whether there are any trade-off which are incurred by the proposed regularizers. In other words, while the proposed method helps when there is a difference in the distribution of labels between the train and validation/test nodes it is not clear how it performs when there is no difference.
* It is not clear whether the highlighted label selection bias is actually present in practice. From the definition of r_i we see that this captures the notion of heterophily, i.e. neighboring nodes have dissimilar labels. In most real-world graphs however, we tend to observe homophily (opposite of heterophily), i.e. neighboring nodes tend to have the same labels. Homophily is often either explicitly or implicitly assumed in many GNN models, so it is not surprising that the performance drops when it is not present. Moreover, it is reasonable to assume that in practice the nodes for labeling are selected uniformly at random (or using active learning) in which case we would likely not observe heavy bias (due to underlying homophily).
* The performance improvement in most cases is marginal and does not seem to effectively mitigate the highlighted issue. In most cases the improvement is between 1% and 2%, and the results for heavy bias are still significantly worse (e.g. >5%) compared to the results for light bias (or no bias, not shown). The two outliers corresponding to 14% and 17% gain might be due to using a fixed data split (see next point).
* The paper uses the Planetoid data splits from [1] to form the validation/test set. Previous work [2] strongly argues against using a fixed split to evaluate the performance of GNNs since considering different splits of the data leads to dramatically different rankings of models. As far as I understood the results in the paper are averaged over 10 random seeds for selecting the training set, but the validation/test set is keep fixed. For a robust evaluation results show be reported as average of several different random validation/test splits.

Question for the authors:
1. How does the results change if we consider the average over a larger number (e.g. 10) of random validation/test splits?
2. What is the performance of GCN/GAT-VD/DVD using uniformly sampled training nodes? Are there any trade-offs?
3. What is the empirical selection bias for standard (uniform sampling) train/validation/test splits, i.e. how large is the difference between the distributions of r_i scores? (For Cora, Citeseer, Pubmed we expect the difference to be small)
4. How well do the proposed methods perform in the transductive setting?
5. How does the performance gain of GCN/GAT-VD/DVD over GCN/GAT on the NELL dataset change as we increase the number of labeled nodes from 1 to some large number?

Additional feedback that did not affect the decision:
* The paper could benefit from a discussion of how the specified notion of label selection bias is similar or different from the notion of homophily/heterophily (see also weak points).
* Another potential selection bias is related to the degree of labeled nodes. For simpler models such as Label Propagation previous work has shown that different variants perform better depending on whether we label high or low degree nodes (see e.g. [3]). It would be interesting to discuss whether this also affects GNNs and whether the proposed approach can help mitigate such bias.
* It would be insightful to investigate whether recent GNN models which can handle heterophily [4, 5] can deal with the label selection bias studied in the paper. The reviewer acknowledges that these papers were made public after the ICLR submission deadline.
* Evaluating the effect of small sample selection bias on massive graphs from the Open Graph Benchmark (https://ogb.stanford.edu/) would be insightful.
* It would be interesting to evalute whether we still observe a strong drop in performance if the training set is chosen in a standard fashion (i.e. the training nodes have high homophily) but the test nodes are selected using e.g. the heavy bias sampling.

## After Rebuttal
Thank you for addressing my questions. Since the performance improvement is still marginal and based on the other reviews I have decided to keep the same score.

References:
1. Yang, Zhilin, William Cohen, and Ruslan Salakhudinov. "Revisiting semi-supervised learning with graph embeddings."
2. Shchur, Oleksandr, Maximilian Mumme, Aleksandar Bojchevski, and Stephan Günnemann. "Pitfalls of graph neural network evaluation."
3. Avrachenkov, Konstantin, Alexey Mishenin, Paulo Gonçalves, and Marina Sokol. "Generalized optimization framework for graph-based semi-supervised learning."
4. Zhu, Jiong, Ryan A. Rossi, Anup Rao, Tung Mai, Nedim Lipka, Nesreen K. Ahmed, and Danai Koutra. "Graph Neural Networks with Heterophily."
5. Zhu, Jiong, Yujun Yan, Lingxiao Zhao, Mark Heimann, Leman Akoglu, and Danai Koutra. "Generalizing graph neural networks beyond homophily."

---

> ### Author Response · Authors · 2020-11-19
> **Response to Reviewer #2 (Part 1/2)**
>
> We appreciate the extensive review and useful comments from this reviewer.  In the last few days, we have made all efforts to carry out more experiments that the reviewer concerns about. Note that due to the time limitation of rebuttal, we only select relative datasets and base models (GCN and GAT) as baselines to respond the corresponding questions.
>
> Q1. What is the empirical selection bias for standard (uniform sampling) train/validation/test splits, i.e. how large is the difference between the distributions of r_i scores?
>
> Thanks for your insightful suggestion to help us to investigate the degree of distribution shift on each dataset more clearly. We calculate the mean value of all r_i for test nodes and also the mean value of all r_i for uniform sampled and biased sampled training nodes respectively.  We summarize the results as follows:
>
> ||Test  |Uniform |Light |Medium|Large|
> | ----| ----  |---- |----|----|---- |
> |Cora  |0.1831| 0.1478|0.2182|0.3118|0.3934|
> | Citeseer|0.2587|0.2975|0.3270|0.4539|0.5651|
> |Pubmed|0.2198|0.2515|0.3457|0.4238|0.4729|
>
> According to the results, the findings are concluded as follows: 1) The mean value of r_i of the test set and the uniform sampled training set have a relatively small gap. It means that there is a slight distribution shift between the test set and the uniform sampled training set. The phenomenon is reasonable because it is hard to achieve unbiased sampling in real-world applications [1]. 2) Training nodes selected with bias have larger gaps of mean value of r_i with test set compared with the uniformly sampled training set. It means that training nodes selected with bias have larger distribution shifts than the uniformly sampled nodes and heavier selection bias can induce larger distribution shifts from the test set. The statistical results indicate that our label selection process can really achieve our goal to make different degrees of distribution shifts.
>
> [1] Huang J, Gretton A, Borgwardt K, et al. Correcting sample selection bias by unlabeled data. NIPS 2007.
>
> Q2.  How does the results change if we consider the average over a larger number (e.g. 10) of random validation/test splits?
>
> Since the reviewer thinks the large gain of GCN-VD/DVD on Citeseer dataset may be due to the fixed validation/test set, we conduct the GCN-VD/DVD and base model GCN on Citeseer dataset over 10 times random validation/test splits. And we report the average Accuracy. The results are shown as follows:
>
> ||Citeseer- Light|Citeseer-Medium|Citeseer-Large|
> | ----| ----  |---- |----|
> |GCN|0.6548|0.5955|0.5224|
> |GCN-VD|0.6929|0.6704 |0.6479|
> |GCN-DVD|0.6945|0.6767|0.6528|
> |Improve.|6.1%|13.6%|24.9 %|
>
> From the results, we get a similar trend as we report in Table 1. The reason for our method achieves larger gain on this dataset may be that the dataset has larger distribution shift than other datasets. The detailed distribution shift statistics of r_i on each dataset could be found in Q1.
>
> Q3. What is the performance of GCN/GAT-VD/DVD using uniformly sampled training nodes? Are there any trade-offs?
>
> We perform GCN/GAT-VD/DVD on three uniformly sampled training nodes datasets (train/val/test split same as [1]) with the inductive setting.
>
> ||Cora|Citeseer|Pubmed|
> | ----| ----  |---- |----|
> |GCN|0.7909|0.7075|0.7845|
> |GCN-VD|0.7980|0.7122|0.7888|
> |GCN-DVD|0.7951|0.7128|0.7874|
> |GAT|0.81|0.7224|0.7714|
> |GAT-VD|0.8133|0.7288|0.7732|
> |GAT-DVD|0.8139|0.7294|0.7735|
>
> GCN/GAT-VD/DVD  also outperform the corresponding baselines. From the statistics of r_i , we know that uniformly sampled training nodes also have a slight distribution shift from test set. (Details in Q1.)  Therefore, the slight distribution shift may be the reason for our model outperforms baselines. In real applications, it is hard to control the collection process without any distribution shift from the training set to the test set [2]. Therefore, we believe our method at least has the same performance as the base model, but we can gain larger improvements with larger bias scenarios.
>
> [1] Yang, Zhilin, William Cohen, and Ruslan Salakhudinov. Revisiting semi-supervised learning with graph embeddings. ICML2016
>
> [2] Huang J, Gretton A, Borgwardt K, et al. Correcting sample selection bias by unlabeled data. NIPS 2007.

---

> > ### Author Response · Authors · 2020-11-19
> > **Response to Reviewer #2 (Part 2/2)**
> >
> > Q4. How well do the proposed methods perform in the transductive setting?
> >
> > We conduct GCN-VD/DVD in transductive setting on Cora dataset with various bias degrees. The following is the results:
> >
> > ||Cora-Light|Cora-Medium|Cora-Heavy|
> > | ----| ----  |---- |----|
> > |GCN|0.7940|0.7906|0.7590|
> > |GCN-VD|0.8074|0.8049|0.7693|
> > |GCN-DVD|0.8061|0.8063|0.7728 |
> >
> > The results indicate our method outperforms the base model consistently in transductive setting. However, it is not  a rigorous way to validate the effectiveness of our method, because the label information may be leaked to test nodes through neighborhood aggregation. Hence the test data is no longer agnostic.
> >
> > Q5. How does the performance gain of GCN/GAT-VD/DVD over GCN/GAT on the NELL dataset as we increase the number of labeled nodes?
> >
> > As NELL is a large-scale graph, we cannot run GAT on a single GPU with 16GB memory. We only perform GCN-VD/DVD and GCN on this dataset.  The number of labeled nodes increases from 1 per class (NELL-1) to 5 per class (NELL-5)  and 10 per class (NELL-10).  The followings are our results.
> >
> > ||NELL-1|NELL-5|NELL-10|
> > | ----| ----  |---- |----|
> > |GCN|0.4416|0.7030|0.7615|
> > |GCN-VD|0.4652|0.7424|0.7734|
> > |GCN-DVD|0.4734|0.7361|0.7727 |
> > |Improve. |7.2% |4.2%|1.5%|
> >
> > From the results, we know GCN-VD/DVD still outperform GCN when the number of label nodes increases,  but improvements on NELL-5 and NELL-10 decrease compared with 1-label case.  The reason is that the distribution shift between training and test could be reduced  when we increase the number of labeled nodes. Moreover, GCN-DVD only achieves competitive results with GCN-VD. One possible reason is that the VD term can already decorrelate variables well when we have enough  labeled nodes, in this scenario, we do not need to differentiate the variable weights.
> >
> > Q6. Discuss the major differences between our method and heterophily methods and whether the highlighted label selection bias is actually present in practice.
> >
> > To our best knowledge, the recently proposed heterophily methods are still based on the assumption that the training set and the test set are drawn from the same distribution. They do not conduct the experiments that training with  heterophily nodes and test with homophily nodes, thus there is no evidence that this kind of methods can be applied to our problem. Moreover,  selecting heterophily nodes is only one kind of scenarios to induce distribution shift. In this paper, we also conduct the experiments on small sample label selection bias. In practice,  selection bias could be induced in many scenarios. For example,  in an image similarity network, where images are connected by their similarity and node attributes are image features,  we may only collect images of dogs on the grass for training while testing with dogs in the water.
> >
> > Thanks for the reviewer's all valuable comments. We will polish the paper in the revision, considering your suggestions.  If you have any other concerns, please let us know.

---

### Official Review · AnonReviewer1 · 2020-10-30
**Important task, unconvincing theory**

**Rating:** 4
**Confidence:** 4

**Review:**

#### Goal

- This work presents an experimental investigation that shows the impact of training selection bias in GNNs (bias with respect to the test data). It also proposes a decorrelation approach to eliminate the spurious correlation in the node representations that come from this training bias.

- I like the experimental investigation (since I think the problem is very relevant). I am skeptical about the method and the results.

#### Quality

- I have many doubts about the validity of the claims.

- Assumption 1 gives us a statistical model not a causal model. In order to claim “causal effects”, one needs to give a structural causal model. The linear models presented later are not linear on the observed variables. “Specifically, for both training and test environment, E(Y∣S = s, V = v) = E(Y∣S = s).” is a covariate shift question, not exactly a counterfactual question if not given with a specific structural causal model.

- “Assumption 2. The true generation process of target variable Y contains not only the linear combination of stable variables S, but also the nonlinear transformation of stable variables.” it is unclear what the authors mean by this statement. That the transformation over S is arbitrary?

- Equation (1): Since the node embeddings can be arbitrary, why do we need an extra function g()? Could that be incorporated into \mathcal{G}(X, A; θg)_S β_S?

- “However, limited by the nonlinear power of GNNs (Xu et al., 2019), it is reasonable to assume that there is a nonlinear term g(G (X, A; θg)_S) ≠ 0 that cannot be fitted by the GNNs.”  This is not at all what (Xu et al., 2019) says. It says that there are some topologies that cannot be represented exactly. While other topologies can be represented exactly. It is entirely dependent on the input data, not a broad general statement for any graph dataset.

- “Hence the parameters of both stable variables and unstable variables would be biased.” This is a strong claim that requires a formal proof.

- The entire procedure is predicated on the linear models of Kuang et al., 2020, since the X in Kuang et al. (in, say, Eq (8) of Kuang et al.) is the observed data (not some learned representation). In this paper, the corresponding variables are H representations obtained by a GNN. The model is no longer linear on the input. The distinction between H_S and H_V is, hence, hypothetical, non-existent, and changes during training, since it depends on the GNN parameters.

- How do we know that the decorrelation of H is not restricted to the training data? How do we know that in the test data, the same decorrelation holds? Any statement of decorrelation carrying over to the test data must be formally shown.

- In section 3.2, all results are for linear models. Then, at some point, the observables X suddenly become hidden H and, it is stated (without proof) that the results carry over?!?!? If this were true, why are most works in the literature limiting their counterfactual evaluations to linear models? I highly doubt one can prove this is true for this scenario.

#### Clarity

- The paper uses very convoluted reasoning to arrive at conclusions that are not at all supported by theory. It uses a lot of results from linear models into a proposed nonlinear model. How can the results in the literature possibly carry over? It is hard to believe any of the claims.

- How can the distinction between S and V be clear in the model, since they are the outputs of a GNN we have not been trained yet?

- Can we precisely define the assumed covariate shift between train and test? No method can work for any covariate shift.

Typos & Overall fixes:

- “Even transfer learning is able to solve the distribution shift problem, however, it still needs the prior of test distribution, which actually cannot be obtained beforehand.” => “Even tough transfer learning is able to solve the distribution shift problem, it still needs the prior of test distribution, which cannot be obtained beforehand”

#### Originality

- Removing GNN training sampling bias with counterfactual inference would be new.

#### Significance

- The task of removing GNN training sampling bias is very important.

#### Pros

- Important task.
- Nice demonstration of the issues with biased training data.

#### Cons

- See “Quality”. I am unconvinced by the method.


----

After rebuttal: My main concerns about (a) no counterfactual model and (b) the linear / nonlinear requirements of the method remain. Generally, bias assumptions are made about the data, not the output of a representation learning procedure.  "The nonlinear relationship between raw input with the outcome can be encoded into the learned embedding" yes, but it does not mean H_S and H_V will meaningfully encode anything related to the input bias in any meaningful way. The method needs to precisely describe the structural causal model to be properly evaluated.

---

> ### Author Response · Authors · 2020-11-19
> **Response to Reviewer #1 (Part 1/3)**
>
> We would like to thank the reviewer very much for the extensive review and useful comments. In the following, we would like to address your comments, hoping they will clarify raised concerns:
>
> Q1. Assumption 1 gives us a statistical model, not a causal model.
>
> Actually, we do not aim to introduce a casual model in Assumption 1.  The assumption aims to assume there is a stable relationship that can be leveraged to stable prediction. And this assumption is not related to the counterfactual question.  We agree  $\mathbb{E}(\mathbf{Y}|\mathbf{S}=s,\mathbf{V}=v)=\mathbb{E}(\mathbf{Y}|\mathbf{S}=s)$ is a covariate shift question, and it means there is an invariant relationship between stable variable $\mathbf{S}$ and outcome $\mathbf{Y}$ in both training and test environments, i.e., $\mathbb{P}(\mathbf{Y_\textrm{train}}|\mathbf{S_\textrm{train}})=\mathbb{P}(\mathbf{Y_\textrm{test}}|\mathbf{S_\textrm{test}})$.  However, the distribution shift between training and test set is mainly induced by the variation in the joint distribution over ($\mathbf{S}$, $\mathbf{V}$) , i.e., $\mathbb{P}(\mathbf{S_\textrm{train}}, \mathbf{V_\textrm{train}})\neq\mathbb{P}(\mathbf{S_\textrm{test}}, \mathbf{V_\textrm{test}})$. Assumption 1 can be guaranteed by $\mathbf{Y}\bot \mathbf{V}|\mathbf{S}$. Thus, one can solve the stable prediction problem by developing a function  $f(\cdot)$ based on $\mathbf{S}$. However, one can hardly identify such variables in GNNs. Therefore, we propose to decorrelate all the variables in GNN.  In fact, we slightly misuse the term “causal effect” and do not intend to give a causal model in this assumption. Thanks for your correction and we will modify our expression in the revision to make it unambiguous.
> Moreover, our model only assumes the linear relationship between the latent variables of raw input that can be learned by the GNN embedding module and outcome, not linearity on the raw input. For more discussion about this question, please refer to Q8.
>
> Q2. The satement of Assumption 2 is unclear.
>
> Assumption 2 attempts to give a general formulation of the relationship between the latent stable variables $\mathbf{S}$ of raw input that can be learned by GNNs and the label $\mathbf{Y}$. Assuming $\mathbf{S}$ is the only input to generate label $\mathbf{Y}$, we believe the real-world graph data is complicated, hence the generation process of $\mathbf{Y}$ not only contains the linear transformation of $\mathbf{S}$ but also the nonlinear transformation of $\mathbf{S}$. Based on the assumption, the generation process can be formulated as:
> $\mathbf{Y}=f(\mathbf{X}, \mathbf{A}) + \varepsilon = \mathscr{G}(\mathbf{X},\mathbf{A};\theta_g)_S\beta_S + \mathscr{G}(\mathbf{X},\mathbf{A};\theta_g)_V\beta_V + g(\mathscr{G}(\mathbf{X},\mathbf{A};\theta_g)_S) + \varepsilon,$
> where $\mathscr{G}(\mathbf{X},\mathbf{A};\theta_g)$ corresponds to an unknown function of $\mathbf{X}$ and $\mathbf{A}$ and it can be learned by a GNN.  $\mathscr{G}(\mathbf{X},\mathbf{A};\theta_g)$ can be composed into stable variables  $\mathscr{G}(\mathbf{X},\mathbf{A};\theta_g)_S$ and unstable variables  $\mathscr{G}(\mathbf{X},\mathbf{A};\theta_g)_V$, where $\mathscr{G}(\mathbf{X},\mathbf{A};\theta_g)_S$ corresponds to $\mathbf{S}$ and $\mathscr{G}(\mathbf{X},\mathbf{A};\theta_g)_V$ corresponds to $\mathbf{V}$ in Assumption 2. As unstable variables  $\mathbf{V}$ do not contribute to the generation of $\mathbf{Y}$, we have $\beta_V$=0.
>
> Q3. Why do we need an extra function g()?
>
> We need the extra function g() based on the following two reasons: 1) Most GNNs only have several nonlinear layers to achieve the optimal prediction results, e.g., 2 layers for GCN and GAT, hence the nonlinear power may be limited. 2) In real-world graph ,  the relationship between  features and topologies with node label is usually extremely complex. Therefore, the true generation process of $\mathbf{Y}$ has a nonlinear term $g(\mathscr{G}(\mathbf{X},\mathbf{A};\theta_g)_S)$ that cannot be learned by existing GNNs and be incorporated into $\mathscr{G}(\mathbf{X},\mathbf{A};\theta_g)_S\beta_S$.

---

> > ### Author Response · Authors · 2020-11-19
> > **Response to Reviewer #1 (Part 2/3)**
> >
> > Q4.  Xu et al., 2019 does not say the limited nonlinear power of GNNs.
> >
> > We agree with Xu et al., 2019 that there are some topologies that cannot be fitted by the GNNs and some can. And  Xu et al., 2019 increase the number of MLP layers in each convolution layer to increase the expressive power of GNNs. Based on this reason, we assume a general form that there is a nonlinear term $g(\mathscr{G}(\mathbf{X},\mathbf{A};\theta_g)_S)$ in the data generation process.  $g(\mathscr{G}(\mathbf{X},\mathbf{A};\theta_g)_S) \neq 0$ means the GNNs cannot fit the relationship between label with features and topologies exactly,  and $g(\mathscr{G}(\mathbf{X},\mathbf{A};\theta_g)_S) = 0$ otherwise. However, in practice, the true relationship is far more complex, thus we believe there is a term $g(\mathscr{G}(\mathbf{X},\mathbf{A};\theta_g)_S) \neq 0$ that cannot be fitted by the GNNs in many graph data. And this paper mainly focuses on this scenario.
> >
> > Q5. “Hence the parameters of both stable variables and unstable variables would be biased.” This is a strong claim that requires a formal proof.
> >
> > Sorry for the confusion. Actually, we have proved the claim following the sentence the reviewer mentioned. As Eq.(3) and Eq.(4) stated, if there is a correlation between $\mathbf{V}$ and $\mathbf{S}$ (or $g(\mathbf{S})$), the estimation of $\tilde{\beta}_S$ and $\tilde{\beta}_V$ would be biased. We have changed the expression of this paragraph in the revision to make the proof more clear.
> >
> > Q6. Compared with Kuang et al., 2020, the GNN model is no longer linear on the input, hence the distinction between $\mathbf{H}_S$ and $\mathbf{H}_V$ is not clear.
> >
> > We propose a general framework，which assumes that there are stable variables $\mathbf{H}_S$ and unstable variables $\mathbf{H}_V$ in node embedding, rather than intending to distinguish these two kinds of variables. In real-world scenarios, the proportion of $\mathbf{H}_S$ and $\mathbf{H}_V$ could be different. In an ideal situation, we only have $\mathbf{H}_S$, then the model could be unbiasedly estimated. Nevertheless, the $\mathbf{H}_V$ is like to exist in many situations, resulting in biased estimation. Meanwhile, the real-world dataset used in Kuang et al., 2020 does not have a strict distinction between the stable variables and unstable variables. We believe this could be a soft concept for variables, which means the latent variable tends to be stable or unstable. And our model and Kuang et al., 2020 both still work in this scenario.
> >
> > Q7. How do we know that the decorrelation of $\mathbf{H}$ is not restricted to the training data?
> >
> > Our model aims to learn a set of parameters $\hat{\beta}$ that could approach the true coefficients $\beta$  in the training stage.  We formulate the GNNs with linear regressor as a linear model on the learned embedding as Eq.(2) stated:
> >
> > $\hat{\mathbf{Y}} = \hat{\mathscr{G}}(\mathbf{X},\mathbf{A};\theta_g)_S\hat{\beta}_S + \hat{\mathscr{G}}(\mathbf{X},\mathbf{A};\theta_g)_V\hat{\beta}_V + \varepsilon.$
> >
> > We partition the GNN model into two parts: the embedding learning part $\hat{\mathscr{G}}(\mathbf{X},\mathbf{A};\theta_g)$ and the linear regression part $\hat{\beta}$. And this is a regular formulation for a GNN with the linear regressor.  In each iteration, we first learn the node embeddings $\hat{\mathscr{G}}(\mathbf{X},\mathbf{A};\theta_g)$, and then fix node embedding and learn the sample weights to decorrelate output of $\hat{\mathscr{G}}(\mathbf{X},\mathbf{A};\theta_g)$, then update the parameters $\hat{\beta}$  and  $\theta_g$.  As theoretically analyzed in Section 2.2, if we decorrelate $\hat{\mathscr{G}}(\mathbf{X},\mathbf{A};\theta_g)$, we would unbiasedly estimate the parameter $\hat{\beta}$. If we could unbiasedly estimate the true effect of learned embedding on label in each iteration, the final estimation would be unbiased, i.e, $\hat{\beta}\approx{\beta}$.
> >
> > Due to the estimated coefficients are equal to the true generation coefficients, we can learn the true relationship between stable variables $\mathbf{S}$ and label $\mathbf{Y}$,  i.e., $\mathbb{P}(\mathbf{Y}|\mathbf{S})$. And this relationship is consistent in both the training graph and the test graph as stated in Assumption 1, thus we can guarantee that the unbiased parameters learned on the training graph can be well generalized to on the test graph.

---

> > > ### Author Response · Authors · 2020-11-19
> > > **Response to Reviewer #1 (Part 3/3)**
> > >
> > > Q8.  In section 3.2, all results are for linear model.
> > >
> > > We propose a general treatment effect estimation framework between the variable $\mathbf{X}$ and $T$ with outcome $\mathbf{Y}$ and assume the linearity between them. However, the $\mathbf{X}$  and $T$ could not only be raw input,  and it can also be the learned embedding. Then we replace the confounder $\mathbf{X}$ with the embedding $\mathbf{H}_\textrm{.-j}$ learned by the GNNs and the treatment $T$ with the embedding $\mathbf{H}_\textrm{.j}$ . The nonlinear relationship between raw input with the outcome can be encoded into the learned embedding $\mathbf{H}$. We have added this explaination in the revision. Thanks for the reviewer's useful comments.
> > >
> > > Q9. Can we precisely define the assumed covariate shift between train and test?
> > >
> > > We define the assumed covariate shift between train and test as following:
> > >
> > > Given a training graph $\mathcal{G}_\textrm{train} =( \mathbf{A_\textrm{train}}, \mathbf{X_\textrm{train}}, \mathbf{Y_\textrm{train}})$ and a test graph $\mathcal{G}_\textrm{test} =(\mathbf{A_\textrm{test}}, \mathbf{X_\textrm{test}}, \mathbf{Y_\textrm{test}})$, and $\mathbf{S}$  and $\mathbf{V}$ represent latent stable and unstable variables in $[\mathbf{A}, \mathbf{X}]$ respectively , in a covariate shift problem,  we have joint distribution $\mathbb{P}(\mathbf{S_\textrm{train}}, \mathbf{V_\textrm{train}}, \mathbf{Y_\textrm{train}})\neq\mathbb{P}(\mathbf{S_\textrm{test}}, \mathbf{V_\textrm{test}}, \mathbf{Y_\textrm{test}})$  and $\mathbb{P}(\mathbf{S_\textrm{train}}, \mathbf{V_\textrm{train}})\neq\mathbb{P}(\mathbf{S_\textrm{test}}, \mathbf{V_\textrm{test}})$,  but conditional distribution between stable variables $\mathbf{S}$ and outcome $\mathbf{Y}$ could be equal in both training graph and test graph  $\mathbb{P}(\mathbf{Y_\textrm{train}}|\mathbf{S_\textrm{train}})=\mathbb{P}(\mathbf{Y_\textrm{test}}|\mathbf{S_\textrm{test}})$.
> > >
> > > Thanks for the reviewer's all valuable comments. We will polish the paper in the revision, considering your suggestions.  If you have any other concerns, please let us know.

---

### Decision · Program_Chairs · 2021-01-07
**Final Decision**

**Decision:**

Reject

**Comment:**

This paper explores a very challenging problem of biased label selection and its effect on graph neural networks. It highlights that GNNs are indeed vulnerable to this issue, and then proposes a regularizer to reduce the learning of spurious correlations from the node embeddings. All of the reviews agree that the problem is relevant and important, but that there are still some outstanding issues.

It’s unclear the degree to which this problem occurs in the real world. It is also important to establish the effectiveness of the method across a range of datasets. The four datasets presented in the paper (and the rebuttal) are a good start, but the reviewers feel that more is still needed to present a convincing argument.

On the theory side, the reviewers are concerned about the linearity assumptions in the theory, and how this will translate into the more realistic nonlinear setting. Even though the authors state that they do not rely on a causal model, the paper and their responses really do seem to point in that direction. This could simply be a clarity issue, in which case I would encourage the authors to revisit this framing this to avoid confusion.

Overall, the paper is promising, but the reviewers feel that more work is needed to provide a comprehensive and convincing case.